# Improved T cell receptor antigen pairing through data-driven filtering of sequencing information from single cells

Helle Rus Povlsen*[†], Amalie Kai Bentzen[†], Mohammad Kadivar, Leon Eyrich Jessen, Sine Reker Hadrup[‡], Morten Nielsen*[‡]

Department of Health Technology at Technical University of Denmark, Kongens Lyngby, Denmark

**Abstract** Novel single-cell-based technologies hold the promise of matching T cell receptor (TCR) sequences with their cognate peptide-MHC recognition motif in a high-throughput manner. Parallel capture of TCR transcripts and peptide-MHC is enabled through the use of reagents labeled with DNA barcodes. However, analysis and annotation of such single-cell sequencing (SCseq) data are challenged by dropout, random noise, and other technical artifacts that must be carefully handled in the downstream processing steps. We here propose a rational, data-driven method termed ITRAP (improved T cell Receptor Antigen Paring) to deal with these challenges, filtering away likely artifacts, and enable the generation of large sets of TCR-pMHC sequence data with a high degree of specificity and sensitivity, thus outputting the most likely pMHC target per T cell. We have validated this approach across 10 different virus-specific T cell responses in 16 healthy donors. Across these samples, we have identified up to 1494 high-confident TCR-pMHC pairs derived from 4135 single cells.

*For correspondence:
herpov@dtu.dk (HRP);
morni@dtu.dk (MN)

[†]These authors contributed
equally to this work
[‡]These authors also contributed
equally to this work

## Editor's evaluation

This paper is of interest to immunologists conducting single-cell analyses of T-cell recognition. It provides improved means of curating datasets to reduce noise and identify T cell-antigen pairs with greater confidence. Experimental data from human virus-specific TCRs are used to validate the methodology.

## Introduction

T cells are essential for immune protection and play a critical role in the immune response to pathogens or cancer, where they directly kill infected or malignant host cells or orchestrate the response of other immune cells. Recognition is mediated through the heterodimeric T-cell receptor (TCR) expressed on the surface of T cells, which engages specifically with a peptide antigen (p) displayed in the MHC. Accurate specificity and broad coverage of antigen recognition are obtained through somatic recombination of the genetic loci, V(D)J, that encodes the α (VJ) and β (VDJ) chains of TCR. The process creates an extensively variable and dynamic repertoire, with an estimated $10^7$ distinct αβTCRs in an individual (*Arstila et al., 1999*; *Davis and Bjorkman, 1988*).

Due to this diversity, the individual TCR transcripts can be used as endogenous cellular barcodes inherited by the T cell progeny. This has been utilized for providing quantitative insight into TCR diversity (*Robins et al., 2009*), to trace lineage decisions of T cells (*Gerlach et al., 2013*) and to monitor the dynamics of T cells across immune-related diseases, such as infectious disease (*Dziubianau et al., 2013*; *Hou et al., 2016*), cancer (*Kirsch et al., 2015*; *Sherwood, 2013*; *Zhang et al., 2018*) and

autoimmunity (*Acha-Orbea et al., 1988*; *Madi et al., 2014*). Most of such TCR repertoire studies have been confined to bulk experiments, tracing the TCR β chain because of its greater diversity (compared to the alpha chain) and because it is less ambiguous due to allelic exclusion (*Bergman, 1999*). However, accurate pairing of the variable TCR α and β regions is valuable for uncovering the biological function of a T cell and is generally lost in bulk experiments since the transcripts are separately encoded. Moreover, we and others have earlier demonstrated that paired α and β TCR data are essential for the characterization and learning of the relationship between the TCR sequence and specificity (*Montemurro et al., 2021*).

To accurately obtain TCR αβ-sequence-pair, single-cell sequencing platforms can be applied to simultaneously capture both TCR chains, while retaining cell origin information. To further assign specificity information to such TCRs, T cells can be stained with barcode-labeled pMHC multimers to simultaneously identify pMHC specificity and TCR sequence of individual cells (*Bentzen et al., 2016*; *Zhang et al., 2018*). Moreover, via DNA barcoded antibodies, the platform facilitates screening of surface proteins to distinguish cellular subtypes and enables cell hashing to trace origin of a given cell to, for example, a given donor, sample, or time point, which is highly valuable in patient studies.

Here we thus applied single-cell sequencing to describe the T cell specificities toward a set of viral-derived peptide-MHC (pMHC) complexes. The pMHCs were selected with the purpose of generating data to expand the current knowledge of TCR-pMHC interactions, and hence covered pMHCs with limited or no paired TCR coverage in the public-domain databases such as IEDB (*Vita et al., 2019*) and VDJdb (*Bagaev et al., 2020*). We deployed the droplet-based single-cell platform from 10x Genomics. Ideally a droplet contains a single cell with all its analytes and a gel-bead in emulsion (GEM). The gel-bead contains barcoded primers that ensures tracing of transcripts back to the cell of origin, referred to as GEMs. While the platform is highly promising, the sequence deconvolution is associated with substantial noise, and challenging to discriminate true from false signals. Common confounding factors include stochastic gene expression, cell cycle variations, apoptosis, and technical artifacts such as multiplet capture, contamination, dropout, and batch effects. Dropout and stochastic gene expression both result in zero-inflated gene counts and are typically insensitive to low expression levels (*Buettner et al., 2015*; *Kharchenko et al., 2014*; *Yamawaki et al., 2021*). Multiplet capture is the event of capturing two or more cells in a single GEM, and it is proportional to the capture rate of cells introduced to the system (*Bloom, 2018*; *Zheng et al., 2017*). The capture rate is determined by the rate of pulsing cells relative to the rate of gel-beads. Thus, to include even low-frequency cell populations, the capture rate must be high at the expense of introducing more multiplets. Contamination is particularly an issue when including analytes such as pMHC multimers, which may be dissolved in cell suspension (*Gaublomme et al., 2019*). The platform has no means of controlling how ambient analytes and their barcodes are partitioned with GEMs, which leads to GEMs that appear like multiplets or consist of ambiguous annotations from multiple analyte barcodes. The reverse issue arises from the risk that analytes may dissociate from the cell before capture. The listed confounders may result in both false-positive and false-negative discoveries.

The main concerns when screening for TCR specificity are nonspecific binding of pMHC and/or cell hashing analytes, incomplete TCR annotation, and T cell multiplets. Nonspecific binding and T cell multiplets may completely dilute the signal from actual interactions, while incomplete TCRs that are missing the annotation for either α- or β-chain render the single-cell setup superfluous. To ensure that a screening is fully exploited and interpreted correctly, we set out to develop a data-driven algorithm that facilitates a consistent and reproducible TCR categorization (clonotyping), peptide-MHC (pMHC) annotation, and antibody-based cell hashing referencing of the donors and their HLA profile.

We applied this algorithm to a dataset derived from screening PBMCs from 16 healthy donors for T cell recognition against common viruses. In total, we evaluated TCR recognition against 10 different pMHC multimers, each labeled with their unique barcode. We demonstrate that following the filtering steps described here we can obtain a confident pairing of pMHC specificity and TCR sequence. This strategy will open novel opportunities to evaluate the structural interplay and the sequence-driven signatures of pMHC recognition at large scale.

## Results

### Parallel capture of TCRαβ sequences, peptide-MHC specificity, and sample origin from single cells

To obtain single-cell-derived triad information on TCR sequence, pMHC specificity, and sample origin, we stained peripheral blood mononuclear cells (PBMCs) from a total of 16 different healthy donors (*Supplementary file 1*). All samples were stained with the same panel composed of 10 different viral-derived pMHC multimers, each labeled with a unique barcode for that specificity and a common fluorescent label (allophycocyanin [APC]) (*Figure 1*; *Supplementary file 2*). We wished to enrich only for TCRs responsive to less commonly reported pMHCs, hence we co-stained the cells with the three most commonly reported viral-derived pMHC multimers (all A0201 restricted: GLC, GIL, NLV) bearing a different fluorochrome (phycoerythrin [PE]) and labeled with their own unique DNA barcode (*Supplementary file 2*). We sorted only the APC-labeled pMHC multimer binding T cells (and hence deselected the PE-labeled T cells) and included these in the downstream single-cell processing.

Prior to sorting, each sample was stained with a distinct hashing antibody to provide a sample identification barcode associated with the GEMs of the resultant single-cell data set. This is done to enable mixing of cells from different samples, while retaining the information of sample origin, and utilizing the capacity of capturing 6000–10,000 cells per lane in the 10x Genomics workflow. This is essential when capturing T cells based on their specificity since the MHC multimer-positive population is generally of low frequency (<1% of CD8 T cells). When several samples are mixed in the process of running the single-cell analysis, all mRNA and DNA barcodes (derived from hashing antibodies or the MHC multimers) associated with a given cell will be encoded with the same 10x barcode, proving the GEM association (*Figure 1*; *Supplementary file 1*).

### Total data from simultaneous capture of cell, TCR, pMHC, and sample ID

The single-cell data is annotated using 10x Chromium Cellranger multi v6.1. This results in each GEM being quantified by a count of unique molecular identifiers (UMIs) (*Kivioja et al., 2011*) for the three components (TCR, pMHC, and sample hashing) based on transcripts of TCR α- and β-chains, barcodes co-attached to pMHC multimers and barcodes co-attached to cell hashing antibodies (*Supplementary file 2*).

To obtain the data presented here, a total of 1800 pMHC multimer-positive cells were sorted per donor irrespective of the frequency or the number of different antigen-specific T cell responses in a given sample, accumulating to a total of 28,800 cells sorted (*Figure 1—figure supplement 1*). All sorted cells were loaded into a single lane for 10x processing. Based on experience with pre-sorting of low frequent cell populations, this equals a total of 6000–9000 captured cells per lane after running the full 10x Genomics 5′ pipeline, and an acceptable doublet rate. This indicates that an appropriate proportion of cells are loaded on the Chromium. Initially, each GEM was annotated based on the most abundant transcripts from TCRαβ, pMHC, and cell hashing. However, this can lead to erroneous annotations as the noise level can differ substantially for the different reagents, resulting in different levels of UMIs.

Based on raw, unfiltered data, we found 6073 GEMs that contained all three components, that is, TCR, pMHC, and sample hashing, corresponding to 40% of the loaded cells (*Figure 2a*). A total of 716,069 GEMs only contained one or two of the components, with the majority containing only the cell hashing barcode (n = 677,502) and the second largest share containing cell hashing as well as pMHC barcodes (n = 37,277). This number vastly exceeds the number of cells in the assay (15,700 cells loaded) and indicates contamination from ambient barcodes in suspension. This is further supported by the observation that the sample hashing UMI count was significantly higher (p<0.0005, Mann–Whitney U) in the 6073 GEMs containing a TCR compared to the GEMs void of TCR (*Figure 2b*). A total of 43,455 GEMs captured a DNA barcode associated with the pMHC library and only 14% of these were completed with TCR transcripts and sample hashing barcodes. In the GEMs containing a TCR, 84% were completed with all three components, that is, included hashing and pMHC barcodes, while less than 0.05% of these GEMs were void of both sample hashing and pMHC barcodes. In the following, we will only consider the 6073 GEMs containing all three components, while taking into account that the high degree of noise also affects these seemingly completely mapped GEMs.

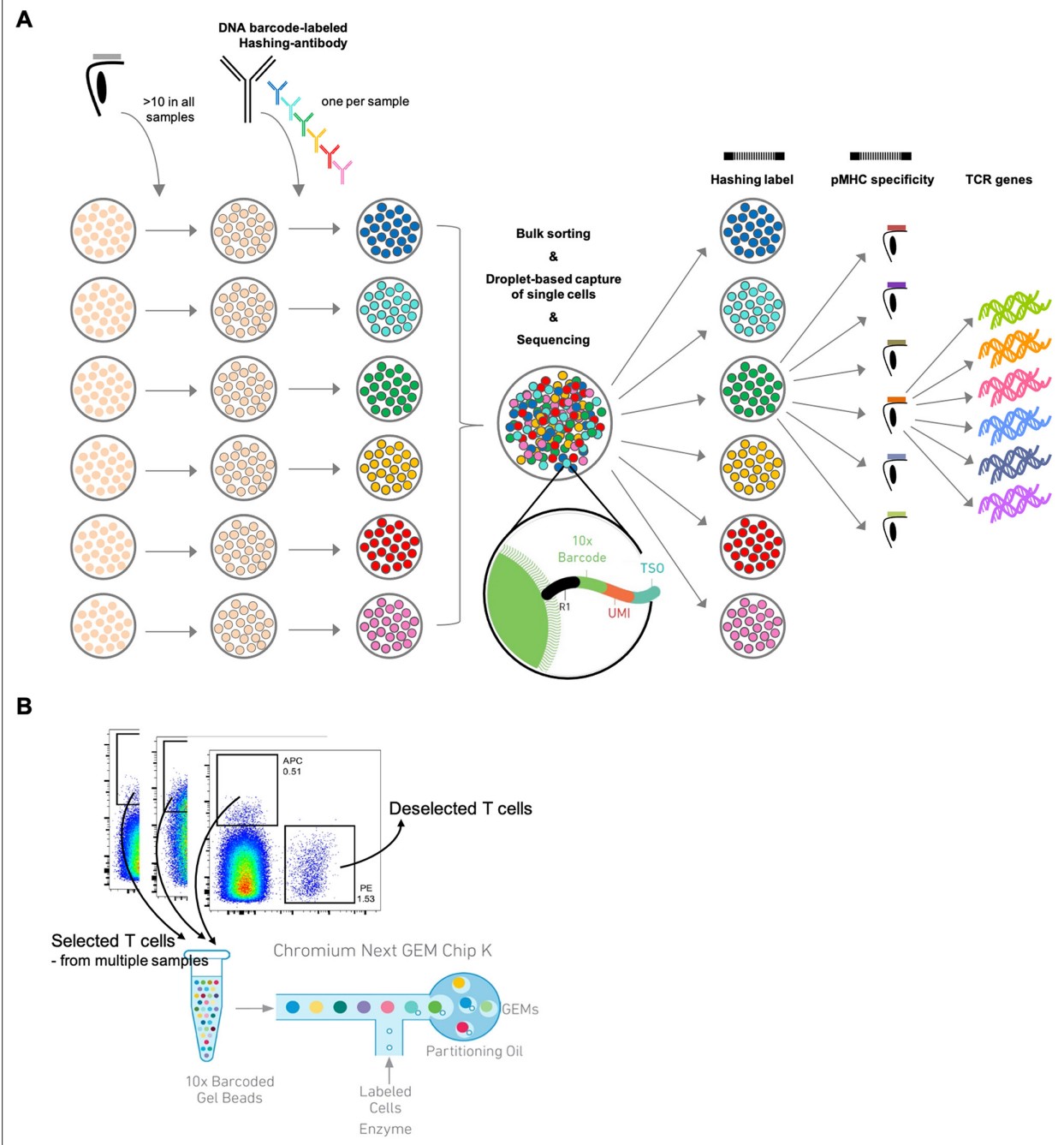

**Figure 1.** Schematic of experimental design. (**A**) Schematic of the experimental strategy. All samples are incubated with the same library of barcode-labeled pMHC multimers and subsequently with a sample-specific barcode-labeled hashing antibody to individually label cells derived from a given sample. Multimer-binding cells from all samples are sorted in bulk and processed through the 10x Chromium workflow. The sequencing output simultaneously captures the sample barcode, the pMHC barcode, and the TCR sequences, which are all matched to a single cell based on the 10x barcode. This also provides the means of retrospectively assigning each cell to their sample of origin via the sample-specific hashing barcode. (**B**) Example showing how the allophycocyanin (APC)-labeled pMHC multimers are sorted collectively from all samples into one tube that is further carried into the 10x workflow. The phycoerythrin (PE)-labeled pMHC multimers are not sorted and hence deselected. A total of 1800 APC-labeled cells are sorted from each donor. Here showing BC126 (large dotplot) and BC341 (small dotplot).

The online version of this article includes the following figure supplement(s) for figure 1:

**Figure supplement 1.** Gating strategy employed for sorting out pMHC binding MHC multimers isolated for single-cell processing.

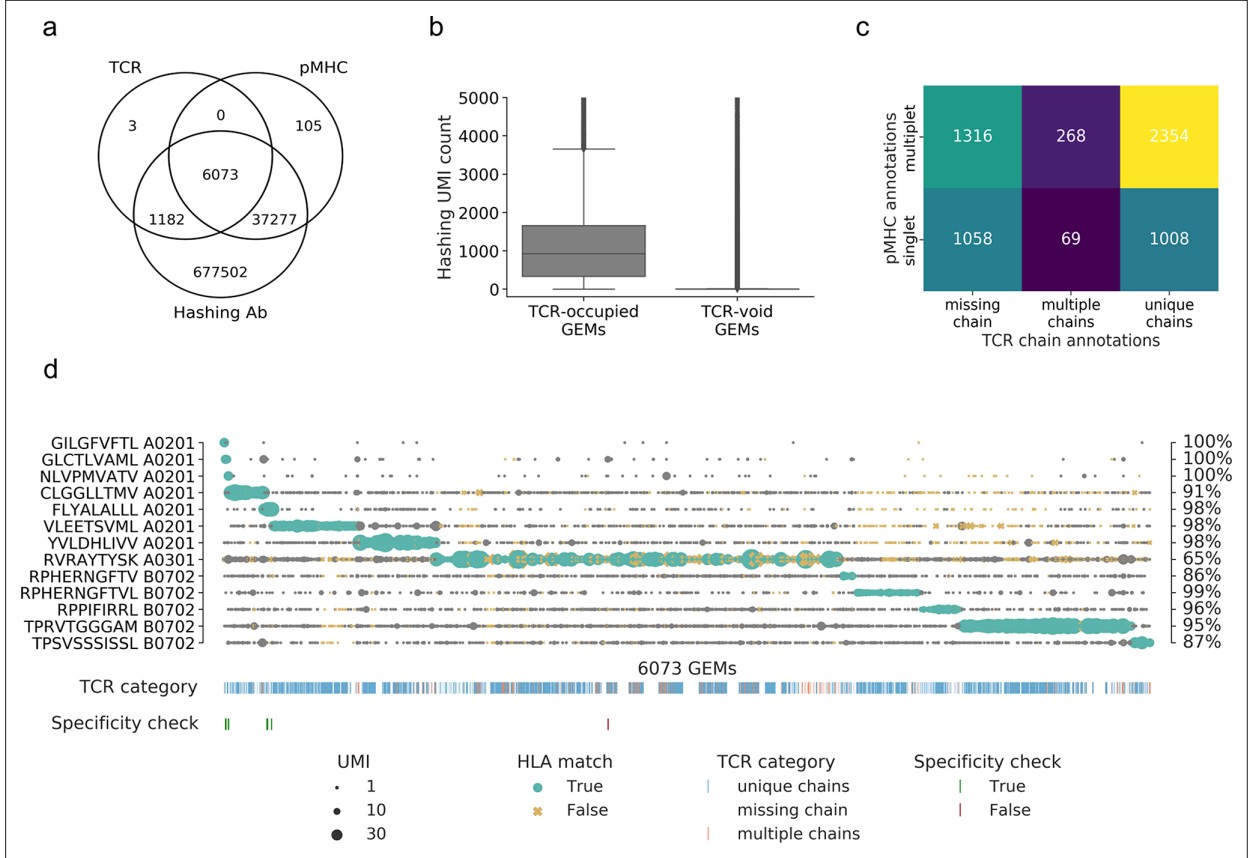

**Figure 2.** Summary of raw data. (**a**) Venn diagram of the content of all gel-beads in emulsion (GEMs) from 10x Chromium drop-seq. Each GEM is expected to contain three components: transcripts of TCR and DNA barcodes from the target pMHC multimer as well as the sample hashing antibody. The Venn diagram illustrates the extent of GEMs with complete capture (capture of all three components) in contrast to the GEMs with incomplete capture (capture of a subset of components). (**b**) Comparison of distributions of unique molecular identifier (UMI) counts of sample hashing barcode between GEMs that contain TCR transcripts (TCR-occupied GEMs) and GEMs that do not contain TCR transcripts (TCR-void GEMs) (p<0.0005, Mann–Whitney U). (**c**) Matrix of the distribution of pMHC singlets and multiplets across GEMs with TCRs either missing a chain, detected with multiple chains, or with a single, unique αβ-pair. The counts are given for each field and illustrated by a color. The lighter color represents higher counts. (**d**) Scatterplot of all detected pMHC barcodes (y-axis) within each of the 6073 GEMs (x-axis) that contain all three components: TCR, pMHC, and sample hashing. In each GEM, the most abundant pMHC is marked with green, while the remaining pMHCs in the GEM are gray. The marker size reports the UMI count of the given pMHC. The marker shape and color recount whether the HLA allele of the pMHC matches the HLA haplotype of the donor, which is deduced from sample hashing (yellow x: non-matching HLA). The fraction of HLA matches within the GEMs displaying a given specificity is annotated to the right of the plot. The first colorbar indicates the type of TCR chain annotation; whether the TCR has a unique αβ-pair, is missing a chain or consists of multiple chains. The second colorbar is a specificity check against the specificity databases IEDB and VDjdb. Colors highlight the GEMs where the CDR3αβ sequences are contained in the databases. The green color represents a match between the database pMHC and the detected pMHC, while red indicates a mismatch.

The GEMs are distributed across three categories of TCR and two categories of pMHC observations: GEMs either missing a TCR chain, contain multiple TCR chains, or contain a unique TCRαβ-pair and GEMs containing either a single or multiple pMHC barcodes (*Figure 2c*). Sample hashing multiplets constitute 100% of GEMs containing sample hashing barcodes, and there is both a large proportion of pMHC multiplets (65%) and GEMs missing either α- or β TCR-chain (39%), hence, multiplets of pMHC and sample hashing is the predominant issue. Few GEMs were detected with multiple TCR α- or β-chains (6%). This may be caused partly by naturally occurring multiplets of α-chain (4%) due to the incomplete gene restriction of the thymocyte during negative selection (*Elliott and Altmann, 1995*; *Petrie et al., 1993*) or due to experimental features of the 1ox platform causing an expected 6.9% of multiplets based on the number of cells loaded in our experiment.

Without further filtering, the pMHC-TCR pairing is subjected to extensive noise (*Figure 2d*), and we capture all the 10 DNA barcodes associated with the APC-labeled pMHCs in a varying number of GEMs. Importantly, the three negative control responses (GIL A0201, GLC A0201, and NLV A0201),

which were present in the donors but not sorted, are only captured in a few GEMs; both as the most abundant pMHC (GIL: 4 GEMs/clonotypes; GLC: 17 GEMs/clonotypes; and NLV: 12 GEMs/clonotypes) and as presumed contamination (i.e., examples where the UMI count of the negative control(s) was not the most abundant). In this latter case, the vast majority (84%) of the negative control pMHCs had UMI counts of 1. Four of the abundant negative control responses matched known IEDB/VDJdb responses. This indicates that the cell isolation via sorting is effective in terms of capturing only the desired cells and relevant pMHC-associated barcode labels. The most frequently detected pMHC across all GEMs is RVR A0301, which is present with high UMI counts across all GEMs. Only RPH(10-mer) B0702-associated UMIs was consistently detected at low numbers per GEM. It was also evaluated whether the HLA allele of the pMHC matches the HLA haplotype of the donor(s) given via cell hashing (*Figure 2d*). Typically, the mismatches are found in GEMs where the most abundant pMHC is detected at low UMI counts while the matches consist of GEMs with higher pMHC UMI counts. Of the 65% GEMs containing pMHC multiplets (*Figure 2c*), 13% contained two or more pMHCs at the exact same UMI level (*Supplementary file 3*), which may either represent noise or true cross-binding events.

The detected specificities in our data have been cross-referenced with the IEDB (*Vita et al., 2019*) and VDJdb (*Bagaev et al., 2020*) databases (*Figure 2d*). Based on the unfiltered data, we found five TCR-pMHC matches (across nine GEMs) and one TCR (one GEM), which was annotated with a different pMHC (*Figure 2d*). This latter is a case of a GEM with multiple pMHCs present with almost equal number of UMIs, where the most abundant pMHC is RVR A0301 (11 UMIs) and the second most abundant pMHC is GLC A0201 (9 UMIs), which is the peptide registered as target in IEDB and VDJdb.

The data in *Figure 2d* suggests that most of the captured T cells interact with several of the screened pMHCs to a degree that exceeds the level expected from natural cross-recognition. Therefore, it is reasonable to assume that a large proportion of these multiplets are formed as a result of ambient pMHC leaking into GEMs.

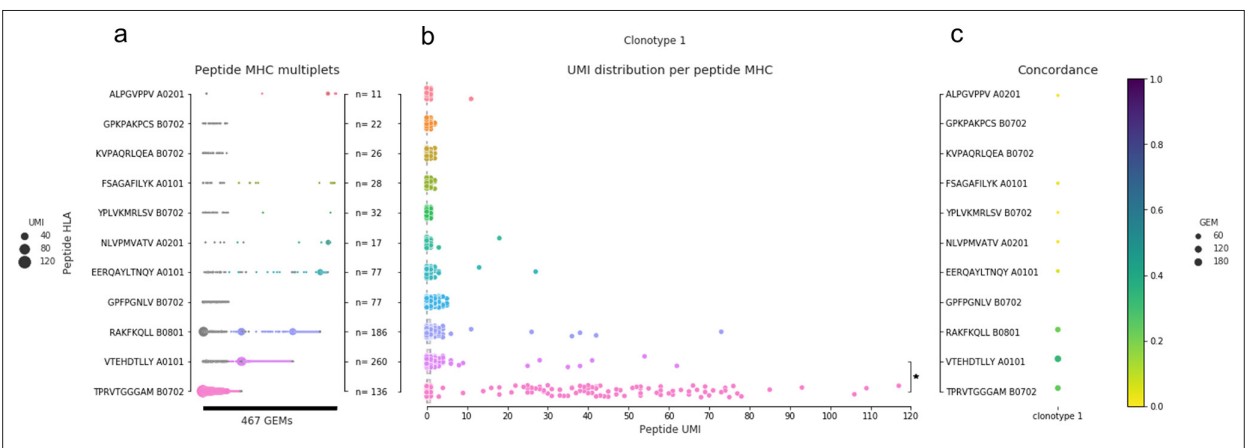

**Figure 3.** An example of pMHC concordance in clonotype 1 (example from pilot study). (**a**) All detected pMHC (y-axis) in each gel-bead in emulsion (GEM) (x-axis, n = 467) of clonotype 1. The marker size shows the unique molecular identifier (UMI) count for the particular pMHC in a given GEM, and the color indicates the pMHC with the highest UMI count, similar to what is shown in *Figure 1d*. If two pMHCs are equally abundant in a GEM, they are both colored. No marker means no detection of that pMHC in that given GEM. (**b**) The compiled distribution of UMI counts for each peptide (assigning 0 UMI when the pMHC is not detected in a GEM). The asterisk marks that a Wilcoxon test showed that the UMI counts of TPR B0702 were on average higher than for VTE A0101 UMI counts. (**c**) The specificity concordance across the GEMs of clonotype 1. Concordance is shown by a color gradient, that is, the larger the fraction of GEMs supporting a given specificity the darker the color.

The online version of this article includes the following figure supplement(s) for figure 3:

**Figure supplement 1.** Clonotype replicas sharing VJ-CDR3ab.

**Figure supplement 2.** Distribution of the three categories of TCR chains across different methods of filtering.

**Figure supplement 3.** Demultiplexing cell hashing using Seurat.

## A data-driven filtering approach

From these observations, it is clear that a substantial part of the data consists of noise, that is, GEMs with multiplets of pMHC and sample hashing, and that the data must be filtered for proper interpretation.

### Clonotype annotation

The definition of T-cell clones (clonotypes) is fundamental for pairing a given TCR clonotype to its respective pMHC recognition. Initial clonotypes were called using 10x Genomics Cellranger, which defines a clonotype as a set of cells that share identical receptor sequences at the nucleotide level, spanning the entirety of the V(D)J-C genes as well as the junction segments. Assuming reliable gene and CDR3 sequence calls by 10x Cellranger, we redefine clonotypes based on TCR annotation. Subsequently, GEMs with no clonotype annotation from 10x were annotated to existing clonotypes conditioned on matching VJαβ-genes and CDR3αβ sequences or as novel clonotypes. Similarly, clonotypes with identical VJ-CDR3αβ were merged to form larger groups of theoretically identical TCRs (*Figure 3—figure supplement 1*). Merging GEMs of the same TCR is essential to make statistical inference based on those groupings, for example, determine expected pMHC target per clonotype. The outcome was a set of 2441 TCR clonotypes across the 6073 GEMs containing both TCR and pMHC. For the 337 GEMs containing TCR chain multiplets, the most abundant chain per GEM was for the subsequent analyses selected to represent the true TCR. Note that this annotation was made post the definition of clonotypes and was applied for the TCR inter- versus intra-similarity comparison.

### Defining pMHC recognition for selected TCR clonotypes

As we have seen earlier, not all GEMs within a given clonotype support the same pMHC target, and defining the pMHC target of a TCR based on individual GEMs thus results in contradicting annotations. The key to identify the expected target for a clonotype is therefore to determine which pMHC identity represents the majority of UMIs across all GEMs within a given clonotype. *Figure 3* illustrates an example from a pilot study that accentuates the importance of studying GEMs in ensemble rather than individually. Most GEMs are annotated with multiplets of pMHCs and across all GEMs the most abundant pMHC varies. While all pMHCs are found most abundant in at least one GEM, three pMHCs (TPR B0702, VTE A0101, and RAK B0801) are more often found most abundant (*Figure 3a*). Although TPR B0702 is detected in fewer GEMs (136) than VTE A0101 (260) and RAK B0801 (186), TPR B0702 is present at generally higher UMI counts (*Figure 3b*). It is evident that there is a difference in UMI distributions between the different pMHC within the GEMs of a given clonotype, and that TPR B0701 is the significantly most abundant pMHC across the ensemble of GEMs even though this pMHC is only present in a minority proportion of the GEM (*Figure 3b*). Based on these observations, we argue that the significantly most abundant pMHC should be annotated as the expected binder for the given clonotype rather than annotating based on the majority.

Having annotated the expected pMHC of a given clonotype, one can next go back to the individual GEMs, and label GEMs where the most abundant pMHC corresponds to the expected binder, as 'true,' and all others as 'false,' and use these annotations to quantify the accuracy of the GEM annotations. Within each clonotype, one can compute a specificity concordance, that is, the fraction of GEMs detected with a certain specificity (defined by most abundant pMHC, i.e., highest pMHC UMI per GEM) (*Figure 3c*). In many cases across the full data set, the expected specificity for a clonotype coincides with the specificity, defined on a per-GEM level, resulting in high concordance. However, for some clonotypes, for example, clonotype 1, GEMs have diverging annotations and therefore lower concordance dispersed across multiple specificities (*Figure 3*). The clonotype visualized in *Figure 3* is specifically chosen to exemplify how this lower concordance can affect the analysis. For clonotype 1, the fraction of GEMs that support VTE A0101 (0.33) is higher than the fraction of GEMs that supports TPR B0702 (0.26). This results in an overall low concordance, and only by considering the complete ensemble of clonotype 1 GEMs can the correct pMHC target be identified (*Figure 3b*).

## Improving concordance between GEM and clonotype annotation based on grid search on UMI features

To rationally filter data, an accuracy metric was defined and optimized through the filtering process. For all specificities belonging to clonotypes with an assigned expected target, we calculated the overall accuracy as the proportion of GEMs where highest abundance pMHC annotation corresponds to the expected target of the clonotype. The raw unfiltered data yielded accuracy and average concordance scores of 69.6 and 83.8%, respectively. Next, we set out to investigate how different data-driven UMI filters could improve these performance values, removing noise and artifacts from the data. This removal would also reduce the number of included observations, hence the performance of different thresholds for filtering the data was evaluated based on a tradeoff between increased accuracy and discarded number of GEMs.

We tested various thresholds on UMI count and UMI ratios, that is, the ratio between the most abundant and second most abundant UMI feature, for pMHC and TCRαβ, respectively. The optimal thresholds were chosen to maximize the weighted average between accuracy and fraction of retained GEMs to favor increase in accuracy above losing some GEMs. This filtering analysis resulted in optimal thresholds of two pMHC UMI counts and a ratio pMHC UMI counts between top one and two >1. The latter results in the removal of GEMs where two pMHC were equally abundant for low UMI counts. The search did not result in thresholds imposing restrictions on neither TCR UMI counts nor TCR UMI ratio, which underpins that the TCRs with a missing chain as well as multiple chains also contribute to good performance. Imposing this filter yielded 4986 GEMs (82% of total), 1494 clonotypes (61% of total), and resulted in 95.3% accuracy, and a mean concordance of 90.6%.

### Additional filters

Additional filters can be added to further clean the data. We investigated how an integrated filter in the 10x Genomics software, Cellranger, performed in removing potential noise from our data set (*Figure 3—figure supplement 2*). The filter (labeled 'is cell') evaluates whether a GEM has captured a cell based on full level of transcriptome data, when available, and otherwise solely on TCR transcript level. The filter was tested with both levels of transcript data, full level and TCR transcript level, which are respectively referred to as 'is cell (GEX)' and 'is cell.' Alternatively, viable cells are identified from the transcript data, independently of Cellranger, based on mitochondrial load and a minimum and maximum gene count per GEM. All three filterings are comparable (*Figure 3—figure supplement 2*) and taken into account in the further evaluations. It is worth noting that, while the filterings based on the full transcript data might remove slightly more noise, the economic costs associated could propose that this should only be applied when the transcript data is required for additional purposes.

Cell hashing is generally a much simpler task to resolve than pMHC multiplets because one hashing entry most often has much higher counts compared to the others (*Figure 3—figure supplement 3*). Moreover, due to the experimental design, where only one hashing antibody is added to each sample, it is expected that only a single hashing signal is associated with each GEM, that is, this does not mirror the complex nature of the pMHC data, where cross-reactivity could result in more than one pMHC be a true binder to a given TCR. Given this simplicity, we opted for utilizing the existing Seurat hashtag oligo (HTO) tool to demultiplex and annotate cell hashing (*Stoeckius et al., 2018*). In this setup, cell hashing also enables filtering based on matching HLA between the donor haplotypes and the HLA of the detected pMHC. Including this additional filter reduces the number of GEMs to 4135 (covering a set of 1494 clonotypes). Additionally, depending on the subsequent use of the data, retaining only complete TCRs containing both α and β may be desirable. Including only GEMs where the TCR-pMHC pair is observed more than once, that is, specificity multiplets, reduces the uncertainty described above. Below we investigate the impact of imposing such filters.

## Impacts of filtering

### Evaluating filters by comparing TCR similarity across specificity

To objectively evaluate the performance impact of the presented filters, we define a quantitative evaluation based on the hypothesis that T cells binding the same pMHC (intra-specificity) will share a higher sequence similarity compared to TCRs of different specificities (inter-specificity) (*Figure 4*). Thus, filtering away artifacts should increase intra-similarity while decreasing the inter-similarity. Here, the similarity score between two TCRs was calculated from the summed score of the pairwise α- and

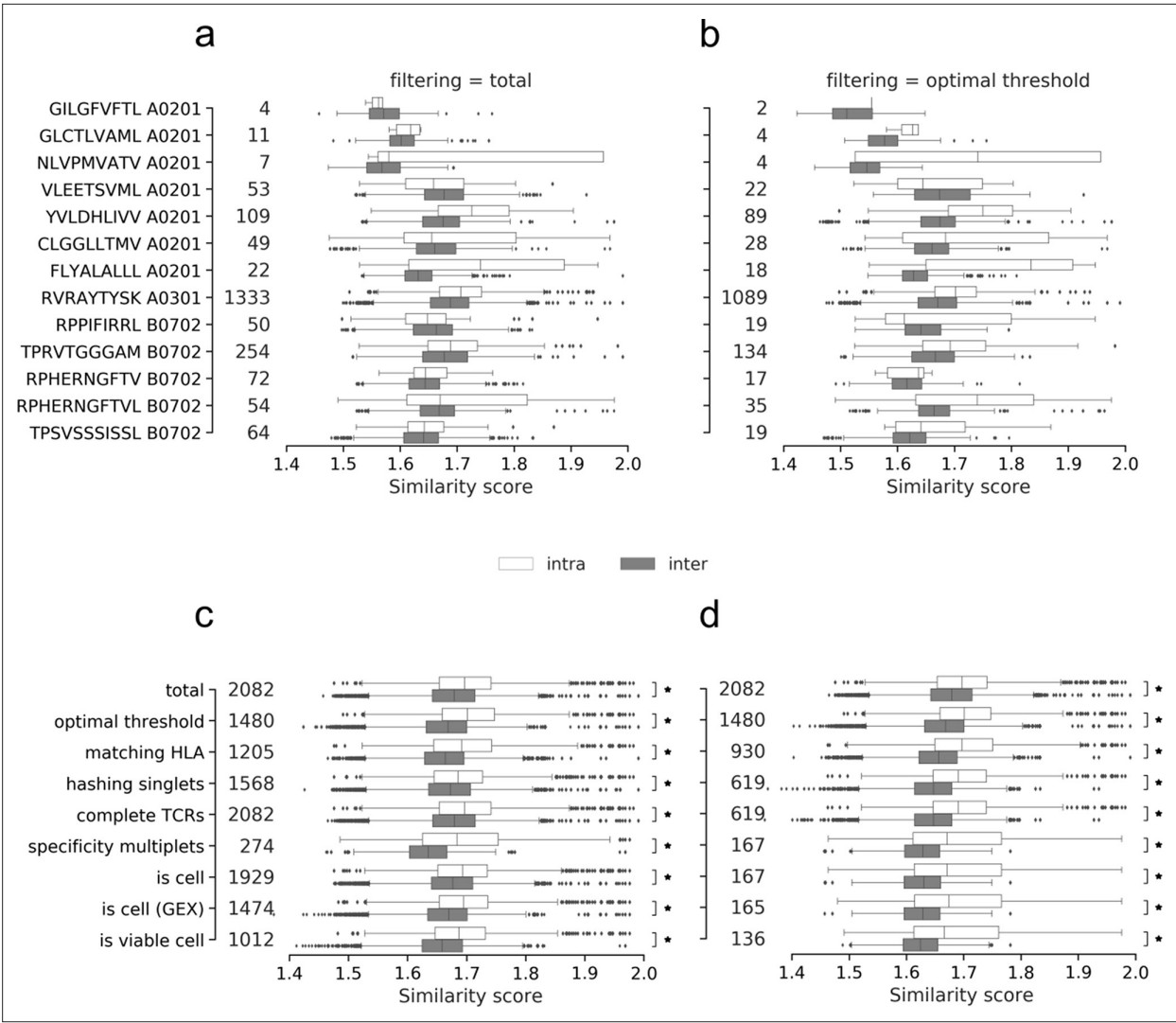

**Figure 4.** Intra- and inter TCR similarity scores per peptide of the (**a**) total (unfiltered) data set and (**b**) the data filtered by the optimized threshold. The similarity per peptide plots (**a** and **b**) illustrates the distribution of paired similarity scores for each clonotype (containing both α- and β-chain). For each pMHC, each clonotype is compared to the remaining clonotypes of the same specificity (intra) and across specificities (inter). The count of compared clonotypes is listed just to the right of the y-axis in both (**a**) and (**b**). (**c**) displays the pooled intra- and inter-scores across all peptides for each of the filtering methods: total (no filtering), optimal threshold, matching HLA, hashing singlets, complete TCRs, specificity multiplets, 'is cell' by cell-flagging, 'is cell' by cell-flagging when including GEX data, and viable cell from analyzing GEX data. An asterisk marks filters where intra-similarity is significantly larger than inter-similarity (Wilcoxon, α = 0.05). (**d**) displays the pooled intra- and inter-scores across all peptides for each of the filtering methods where each filtering is added cumulatively to the previously listed above it. An asterisk marks filters where intra-similarity is significantly larger than inter-similarity (Wilcoxon, α = 0.05). The count of compared clonotypes is listed just to the right of the y-axis in both (**c**) and (**d**).

β-chain similarities calculated using a kernel method described in *Shen et al., 2012* and applied in *Chronister et al., 2021*.

Based on this kernel similarity metric, the filters were tested individually and cumulatively, that is, each filter was added to the previous set of filters. The general trend is that TCRs with the same specificity are more similar to each other than to TCRs of different specificities, when computing the intra- and inter-similarities per pMHC before and after filtering on the optimized UMI thresholds (*Figure 4a and b*). Before filtering, 9 out of 13 pMHCs displayed a higher mean intra-similarity than inter-similarity scores, whereas this number was 11 out 13 pMHCs when applying the UMI thresholds. The outliers before filtering were GIL A0201, VLE A0201, CLG A0201, and RPP B0702, while the outliers were reduced to VLE and RPP after filtering. Generally, the similarity scores often have a wide, overlapping range between the intra- and inter-categories. The three pMHCs that were deselected during sorting,

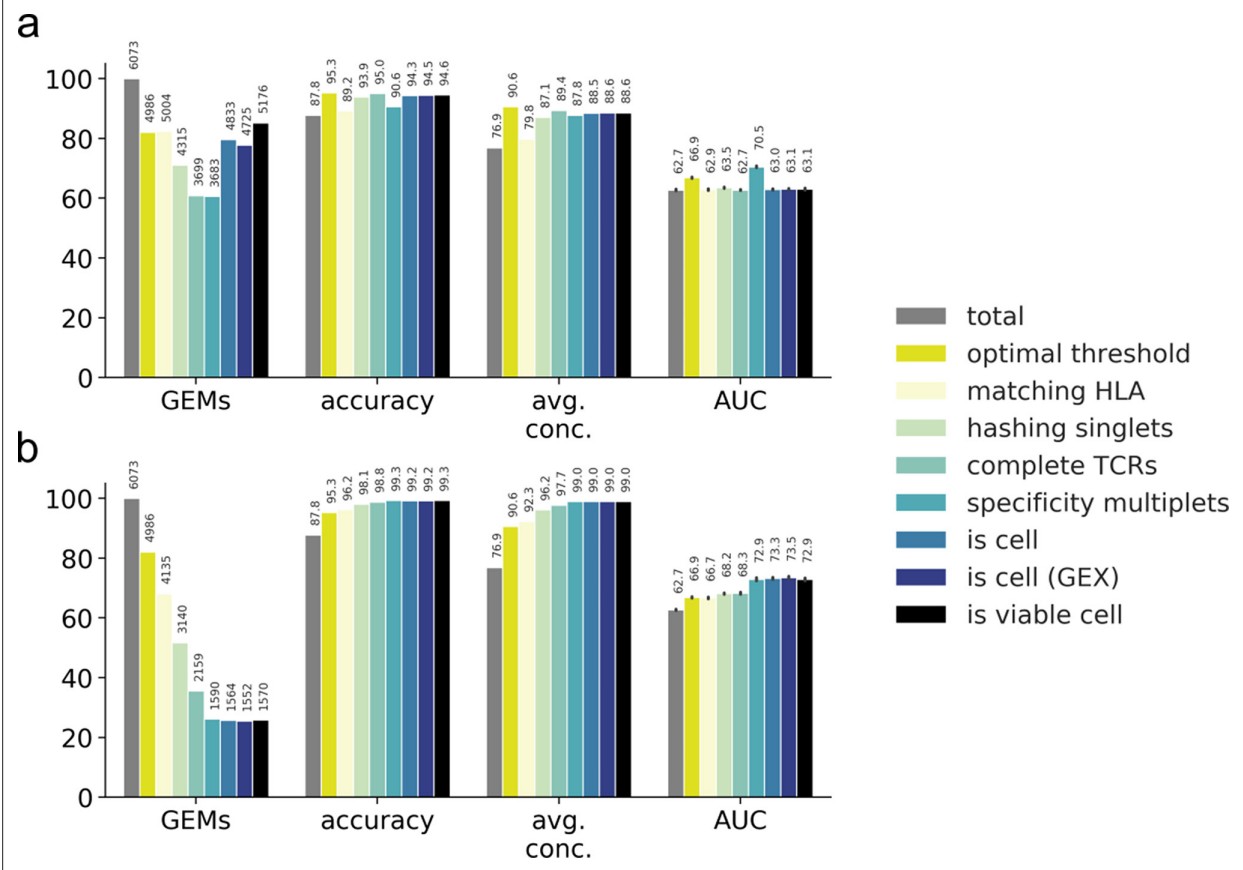

**Figure 5.** Performance metrics for evaluating the filtering steps. Performance is measured by number and ratio of retained gel-beads in emulsion (GEMs), accuracy defined by proportion of GEMs where most abundant pMHC matches the expected binder (accuracy), average binding concordance (avg. conc.), and AUC of similarity scores (AUC). The filtering steps consist of total (raw, unfiltered data), optimal threshold obtained from grid search, matching HLA, hashing singlets identified from Seurat HTO demultiplexing, complete TCRs with a unique set of α- and β-chain, specificity multiplets such that each TCR-pMHC pair must be observed in two or more GEMs, is cell defined by 10x Genomics Cellranger, is cell (GEX) defined by Cellranger where GEX data is included, and is viable cell defined by mitochondrial load and gene counts. (**a**) Presentation of the individual effect of each filter. (**b**) Presentation of the accumulated effects of the listed filters.

The online version of this article includes the following figure supplement(s) for figure 5:

**Figure supplement 1.** Overview of ITRAP pipeline steps: ITRAP core actions are colored blue, while supporting steps are colored gray.

GIL A0201, GLC A0201, and NLV A0201, are only detected in a few TCR binding events. To enhance the power of comparison, the intra- and inter-scores were pooled respectively across the individual pMHCs (*Figure 4c and d*). The results demonstrate that intra-similarity is significantly higher than inter-similarity at each filtering step, both individually and combined. Moreover, we observe that the differences between intra- and inter-similarity appear to increase as filters are cumulatively added and fewer observations are left (*Figure 4d*). Particularly, the median inter-similarity score is lowered, suggesting that the filtering steps predominantly remove false-positives.

## Evaluating filters across selected performance metrics

To compare the effect of the filters, the similarity scores were converted to the performance metric: AUC (area under the receiver operating characteristic [ROC] curve). Here, intra-specificity comparisons are regarded as true-positive observations and inter-specificity comparisons as true-negatives. Based on these performance metric definitions, we quantify the effect of each filtering step (*Figure 5*) and find that the highest accuracy and highest average concordance are obtained by filtering on the optimal threshold (95.3 and 90.6%), while the highest AUC is obtained from filtering on specificity singlets (70.5%) (*Figure 5a*). Expectantly, the accuracy and average concordance increase when the filters are imposed cumulatively (*Figure 5b*). The accumulation of filters also results in drastic

reduction of the GEMs, and it is evident that one must carefully weigh out the need for specificity over sensitivity when selecting the desired set of filters.

We conclude that the minimal filtering must include optimal threshold and matching HLA between pMHC and donor haplotype. Filtering on specificity multiplets would inherently result in more reliable observations, risking the removal of rare, low-avidity binding events. Generally, we did not find that including GEX data improved performance considerably. Finally, filtering on incomplete TCRs yields the second highest accuracy and average concordance. Unfortunately, the filter almost halves the number of GEMs. Hence, this filtering should be considered depending on future use of the data. An overview of the pipeline can be found in *Figure 5—figure supplement 1*.

## Inspecting the filtered data

To determine the impact of the filtering steps, we have compiled the binding concordance for all clonotypes and applied three selected filtering steps: (1) the raw, unfiltered data, (2) filtering on optimal UMI thresholds and matching HLA, and (3) additionally filtering on complete TCRs (*Figure 6*). The raw, unfiltered data displays many clonotypes where the most abundant pMHC in GEMs of a given clonotype are dispersed across multiple of the screened pMHCs (*Figure 6a*). When imposing the recommended set of filters, optimal threshold, and HLA match, the outliers are greatly reduced, although not all low-concordance GEMs are removed (*Figure 6b*). By additionally filtering on complete TCRs, even fewer outliers are left (*Figure 6c*). Note again that we have purposely deselected T cells specific for GIL A0201, GLC A0201, and NLV A0201, explaining the few observations for these otherwise frequently recognized epitopes. An overview of gene usage for clonotypes specific for each of the 10 positive pMHC can be found in *Figure 6—figure supplement 1*.

Many of the remaining low-concordance GEMs still suggest the improbable event of cross-binding across HLA restriction. We suspect that these are artifacts that we have not successfully removed. When the most strict filtering is imposed (*Figure 6c*), there are 77 GEMs (out of 2833) with a binding concordance of 0.5 or lower, which will be referred to as outliers. Also, 72 of those GEMs contain pMHC multiplets. And 93% of the multiplet outliers actually do contain the pMHC, which defines the high-concordance GEMs, however, at a lower UMI count. In the GEMs with multiple pMHC annotations, the HLA is conserved across the pMHCs in 7% of the cases. In 82% of the cases, the HLAs are different, but still match the HLA haplotype of the donor given by the cell hashing. Of the 77 outliers, the most dominant pMHCs are RVR A0301 (38%) and TPR B0702 (30%). These values are in line with the overall contribution of these pMHCs in the GEMs with a binding concordance above 0.5 (34 and 21%). This suggests that the observed GEMs with low concordance are a result of noise and not a reflection of TCR cross-reactivity. Prior to filtering the data, six clonotypes were identified, which were already registered in IEDB and VDJdb, five with matching pMHC, and one with a different annotation than in our observation (*Figure 2d*). The five matching clonotypes (nine GEMs) were successfully retained, while the mismatching clonotype (one GEM) was filtered away. Four of these nine GEMs were negative control responses (two GLC and two NLV) and the remaining five GEMs (across two clonotypes) were all responses directed towards FLYALALLL A*02:01. All pMHCs were detected with UMI counts ranging from 17 to 46.

## Comparing single-cell data with fluorescent-based pMHC multimer screening

### Investigating dominant clones

Beyond mapping the landscape of known TCR-pMHC interactions, single-cell screening enables investigation of T cell repertoire diversity. The high resolution both reveals the specificity and the TCR clonality within the individual T cell populations, which is not possible to recover in classical stainings using fluorescent-labeled pMHC multimers (fluorescent multimers). The T cell diversity in the nine donors toward the set of analyzed pMHCs reveals a clear hierarchy with dominant responses in fluorescent multimer staining (*Figure 7a*); however, the clonality of each specificity is only available via single-cell data (*Figure 7b*). Here ITRAP represents data filtered by optimal UMI thresholds and matching HLA between pMHC and donor haplotype (given via cell hashing). Single-cell screening further enables comparison of the clonal distribution and the total clonal size per specificity. In this respect, the samples BC328 and BC62 are strikingly similar in their distribution of expanded clones. They both display a large and broad response toward RVR A0301 and two smaller responses toward RPP B0702

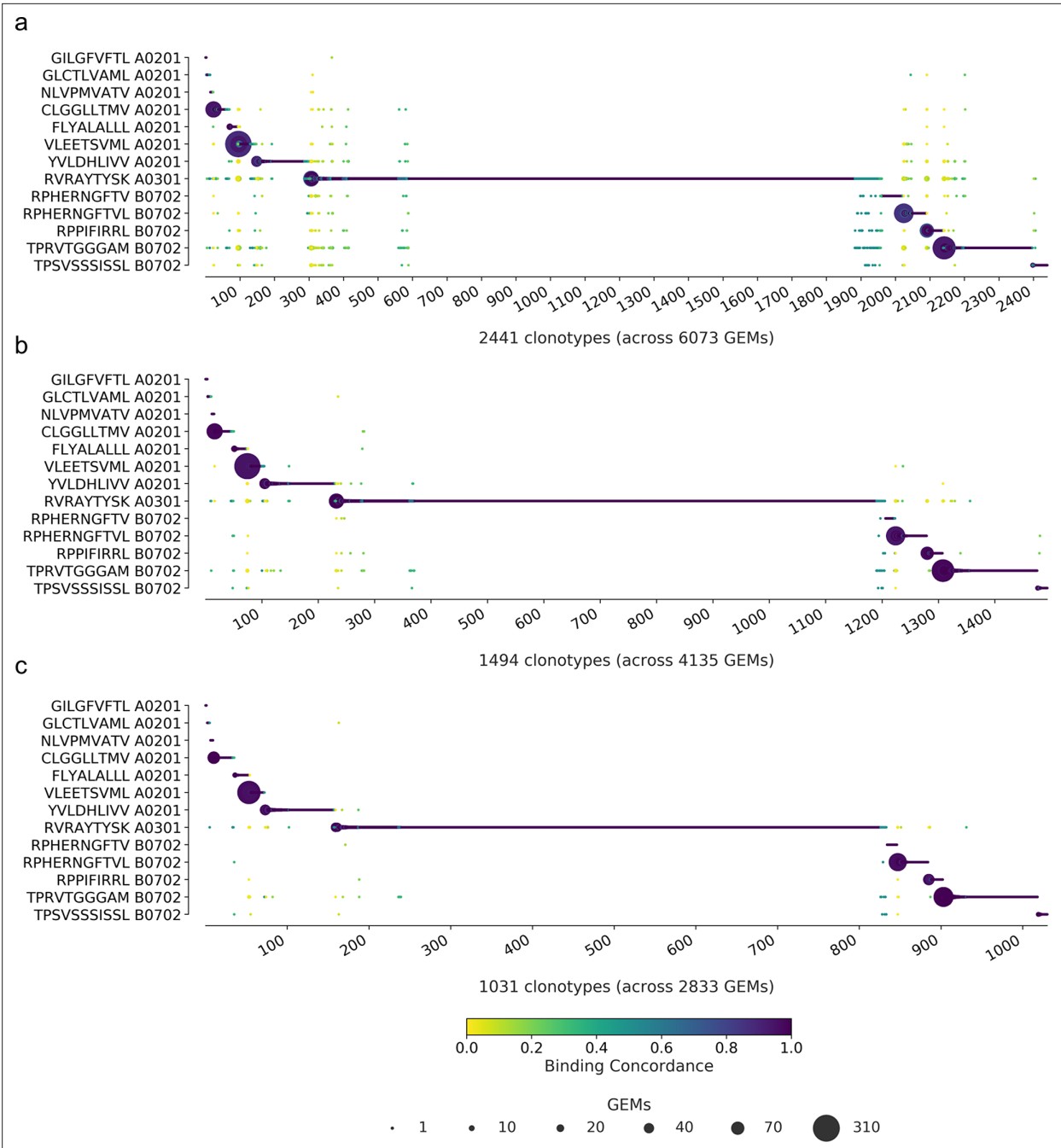

**Figure 6.** Specificity per clonotype. The library peptides are listed on the y-axis and each clonotype is represented on the x-axis. Below the x-axis is annotated the total number of clonotypes and gel-beads in emulsion (GEMs) in the presented data. The marker size shows the number of GEMs supporting a given specificity. The color indicates the binding concordance that is calculated as the fraction of GEMs within a clonotype that support a given pMHC. The higher the concordance, the larger the fraction of supporting GEMs. The three plots illustrate the impact of three filtering criteria. (a) presents raw data with no filtering applied. (b) presents data filtered on optimal threshold and HLA matches. (c) presents data filtered as in (b) with the additional requirement of only complete TCRs (note that cell hasting filtering was not included here). A summary of the specificity singlet distribution for each subfigure is presented in *Supplementary file 5*.

The online version of this article includes the following figure supplement(s) for figure 6:

**Figure supplement 1.** Gene-gene pairing of V and J gene segment usage for each TCR chain.

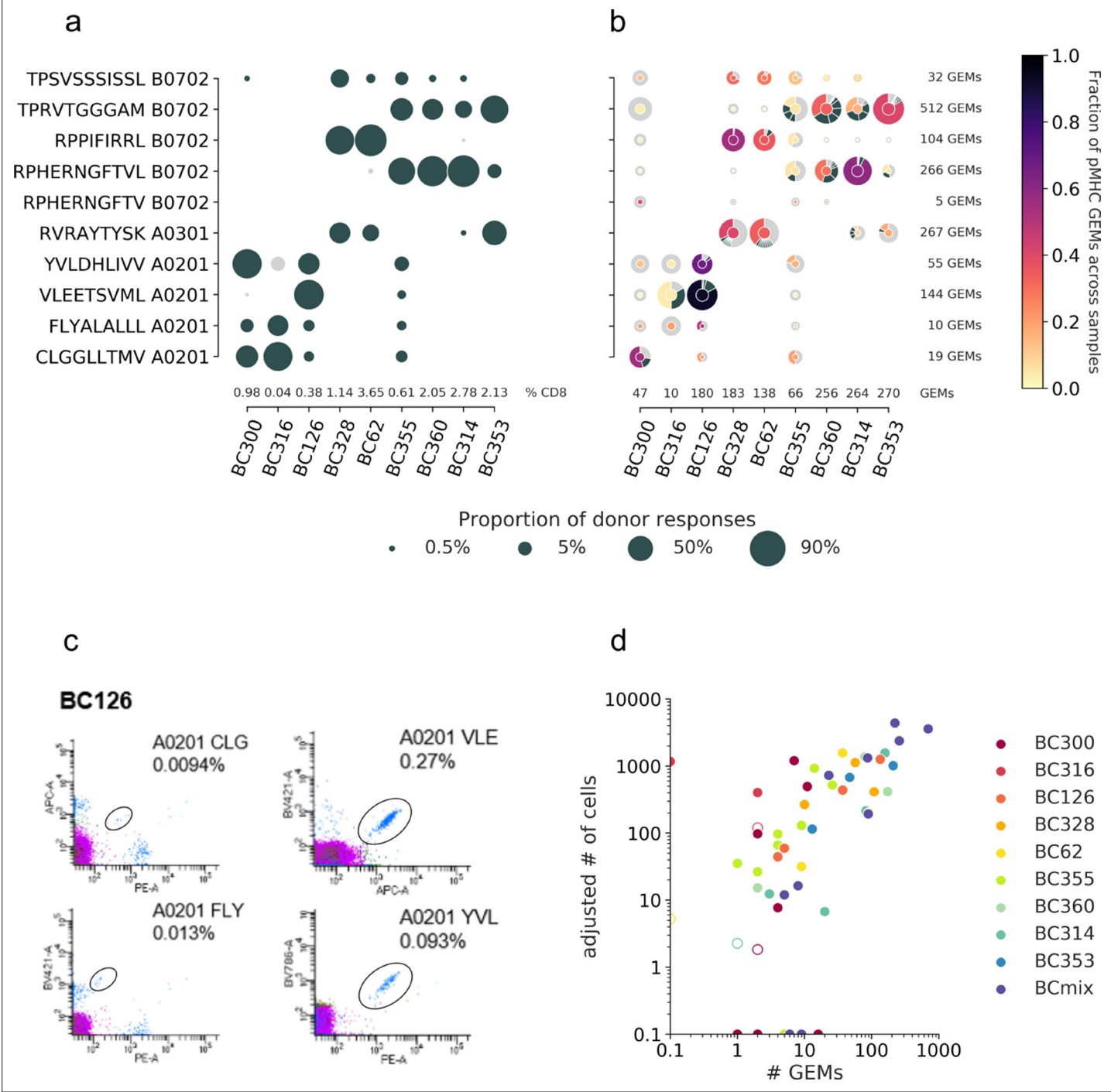

**Figure 7.** T cell diversity per peptide across the individual samples. The nine samples, PBMCs from nine individual donors are represented on the x-axis. The marker size defines the distribution of T cells recognizing a given peptide, normalized per sample. (**a**) The T cell frequencies are visualized as the proportion of a given multimer-positive response within a donor. The black markers represent responses detected above the threshold, that is, ≥10 cells and ≥0.002% of total CD8 T cells, or ≤10 cells but ≥0.01% of total CD8 T cells. The gray dots represent detected specificities below threshold but represented by ≥2 cells. Summed frequencies of detected responses within a donor are given as % of total CD8 T cells and listed just above the x-axis. (**b**) The T cell frequencies are based on gel-bead in emulsion (GEM) counts normalized per sample from the single-cell data. Absolute GEM counts per sample are listed above the x-axis. The marker is colored by the fraction of GEMs within a specificity that originate from a given sample. Absolute GEM counts per peptide are listed to the right of the plot. The marker contains a donut diagram illustrating the distribution of clonotypes specific for the given peptide in the given sample. The wedge that represents the dominant clone is colored according to the center of the donut. Remaining clones (>1 GEM) are anthracite gray, and all clonotypes only supported by one GEM only are pooled and represented by a single light gray wedge.

*Figure 7 continued on next page*

*Figure 7 continued*

Comparing the sizes of the T cell populations for each specificity per donor between the two screening methods in (**a**) and (**b**) yielded the following Spearman correlations: BC126 (1.00, p<0.0005), BC328 (0.90, p=0.006), BC355 (0.74, p=0.02), BC360 (0.89, p=0.04), BC314 (0.90, p=0.04), and BC353 (1.00, p<0.0005). (**c**) Representative example showing the four different responses detected with fluorescent-labeled pMHC multimers in donor BC126. (**d**) Correlation between T cell responses detected by fluorescent-labeled MHC multimers (y-axis) and single-cell capturing (x-axis). Correlation is given by Pearson correlation coefficient 0.73 (p<0.0005). The responses from fluorescent-based screening are given as an adjusted number of cells based on the detected response frequency out of 1800 cells (see calculations in *Supplementary file 4*). The hollow markers represent responses below detection threshold as described in (**a**). The responses are colored by the donor-of-origin. BC mix corresponds to BC311, BC11, BC83, BC88, BC341, BC342, and BC76.

and TPS B0702, which are both dominated by a single clonotype. Further, most peptides elicit diverse relative responses between samples. For example, RPP B0702 is the dominant response in samples BC328 and BC62, but the minority response in sample BC314. Sample BC300 contains primarily small clones, that is, fewer cells in each clonotype; however, this sample is generally represented with low amounts of total data (46 GEMs). Of note, small clones might be a result of suboptimal single-cell capture or because high-frequency responses can potentially mask any lower frequency responses present in a given donor (*Supplementary file 4*) when only 1800 cells are sorted from each sample. Samples represented with many GEMs are expected to be fully covered and therefore may contain more different expanded clonotypes, as sample BC360.

## Evaluating ITRAP by 'ground truth' of fluorescent-based pMHC multimer screening

The net overlap of identified T cell responses between the two screenings (*Figure 7a and b*) is estimated to 0.63 by Matthew's correlation coefficient (MCC). Most of the T cell populations detected by fluorescent multimers are also captured in the single-cell screening, reflected by a recall of 0.95. However, the single-cell capture of small T cell clones (*Figure 7b*) that were not detected using fluorescent multimers (*Figure 7a*) negatively impacts the precision, yielding a score of 0.71. For fluorescent-labeled MHC multimers, a detection threshold of 10 events is applied, which may account for the difference observed for the very rare clone, of which most were only represented by one GEM per clonotype (demonstrated by a light gray outer circle in *Figure 7b*). In only two cases, T cell populations were detected with fluorescent multimers but not captured in single cells: BC316/CLG A0201 and BC62/RPH(10mer) B0702 (*Figure 7a*). The large T cell population of BC316/CLG A0201 was likely a technical artifact related to the barcode-labeled pMHC multimers.

We calculated the number of antigen-specific T cells sorted per donor, based on the total number of sorted cells/donor (n = 1800) and the frequency of each T cell population (*Figure 7c* and *Supplementary file 4*). This number of sorted cells for a given specificity was strongly correlated with the numbers of single-cell GEMs assigned to the same specificity (Pearson correlation coefficient, PCC = 0.73, p<0.0005). Hence, even though many of the TCR-pMHC pairs are found across low numbers of GEMs, the strong correlation indicates that we can confidently assign pMHC specificity to the TCR sequence using ITRAP even in these cases. This is an essential feature of such analyses tool since individual antigen-specific T cell responses are often present at very low frequencies and patient material is often scarce. Hence, to conduct biologically relevant studies, strategies that enable the investigation of the breadth of antigen-specific T cell responses are required. We also fitted a linear regression for T cell populations sorted and assigned with at least one adjusted cell count or GEM in the log-log space (R2 = 0.56). The regression indicates that ~10% of sorted cells will be captured in a single-cell screening with TCR-pMHC information yielded by ITRAP.

## Discussion

Here, we have described and validated ITRAP, a data-driven approach for Improved Pairing of T cell Receptor and Antigen. We have successfully filtered single-cell 10x Genomics data to identify reliable TCR-pMHC interactions of up to 1494 clonotypes. The method can be adapted to any single-cell immune profiling data set and is highly transparent in the steps taken, allowing the user to choose appropriate stringency of filtering.

Our recommended approach of cleaning data with minimal elimination of GEMs is obtained by two sets of filters: (1) the optimized data-driven UMI thresholds combined with (2) information on matching HLA specificity (as obtained from donor-specific hashing). Increasing filtering is naturally at the expense of the number of GEMs that might reflect the trade-off between specificity and sensitivity of the assay. However, any benchmarking or validation is made difficult without a golden standard. Our best attempt at quantifying the impact of filtering is based on three metrics: annotation accuracy, binding concordance, and AUC of clonotype similarity for which ITRAP yielded the scores 96.2, 92.3, and 66.7, respectively. Evaluation of ITRAP with responses from fluorescent pMHC multimer staining revealed strong correlation (PCC = 0.73, MCC = 0.63) between the number of sorted T cells and the number of detected GEMs across all specificities.

Accuracy of pMHC annotation was based on selected clonotypes where the expected target was statistically distinct and UMI thresholds were set to optimize the annotation accuracy. Rare clonotypes are not considered in this metric and clones are not expected to display cross-reactivity amongst the included pMHC multimers. The optimal UMI thresholds are intended to remove observations deviating from the expected target. The thresholds are based on the assumption that contamination will predominantly exist at lower UMI counts than actual binding events. This limits the sensitivity of the method in cases of low-affinity low-frequency interactions that otherwise might be of great scientific and clinical interest. It is, however, essential to underline that ITRAP does not explicitly exclude cross-reactivity, and cross-reactive events are maintained after threshold filtering if such GEMs exist with proper UMI count values. This is in contrast with other tools where cross-reactive events are explicitly excluded.

The binding concordance is a metric that highlights cross-reactive clonotypes. In assays where cross-reactivity is not an expected outcome, binding concordance can be useful to evaluate the clonotypes where an expected target could not be identified. On the contrary, for data where T cell cross-recognition is of particular interest, the binding concordance can be used to establish the relative TCR binding contribution of each of the attributed pMHC targets. Growing evidence points to the relevance of T cell cross-recognition in both infectious disease (*Dowell et al., 2022*) and cancer (*Fluckiger et al., 2020*). Hence, novel tools to interrogate this phenomena on a single T cell level are highly warranted.

The last evaluation metric, AUC of clonotype similarity, is based on the assumption that T cells sharing specificity have more similar TCR sequences than T cells of different specificities (*Chronister et al., 2021*). This approach showed increasing separation of intra-specificities and inter-specificities as filters were cumulatively added, indicating that nonspecific binders were effectively removed. To further increase the AUC, discarding clonotype singlets (i.e., TCR clonotypes represented by only one GEM) was the best single filtering step to improve the AUC of similarity scores (AUC = 70.5, *Figure 5a*). This likely reflects that a fraction of such clonotype singlets represents nonspecific binding events. However, removing these as a standard procedure of ITRAP results in a substantial loss of TCR capture, represented by all T cell specificities with a light gray outer circle in *Figure 7b*. Thus, when aiming for capture of very low-frequency T cell specificities, a balance should be made between including this more stringent filtering step, or including such events, as demonstrated here.

The UMI thresholds identified by ITRAP are data specific and cannot be universally applied but must be fitted for individual experiments. To investigate this and the robustness of ITRAP further, we have in a recent publication (*Povlsen et al., 2023*) applied the method to the large 10x Genomics data set available at https://support.10xgenomics.com/single-cell-vdj/datasets. In short, we find that the ITRAP method also for this data set was capable of accurately denoising the raw data. This accuracy was manifested not only through highly improved internal performance (concordance, accuracy, and AUC) metric values, but also in improved predictive power of a NetTCR (*Montemurro et al., 2022*) machine learning method trained on the denoised data. Due to the very different experimental settings in the two data sets, the obtained UMI-based filtering thresholds as expected varied between the two studies. This indicates the high strength of ITRAP being robust at being applied to data generated in very different biological and technical settings.

Other than investigating the robustness of ITRAP, the paper also benchmarked ITRAP performance against ICON (Integrative COntext-specific Normalization) (*Zhang et al., 2021*). ICON was similarly developed for denoising and discriminating true TCR-pMHC binding signal from nonspecific background noise in single-cell data. The results from this benchmark suggested an overall superior

performance of the ITRAP method compared with the ICON method in terms of both data consistency and performance.

ICON was developed based on the public 10x Genomics data that includes 6 negative control pMHCs (*Boutet et al., 2019*) and 44 pMHC for positive selection of T cell populations. Comparing our method with ICON suggests that we present a more flexible and customizable approach. Where ICON yields a specific data set (*Zhang et al., 2021*), the ITRAP method allows varying yields, depending on the level of filtering applied. The ITRAP method requires haplotype information for optimal filtering, which ICON does not consider resulting in ~15% mismatched HLAs of their specificity annotations. Further, we believe that negative control pMHCs may provide false confidence as we do not know the limits to T cell specificity. For both methods, a particular awareness should be assigned to properly handle the range of affinity displayed by different clonotypes. One clonotype may display natural low affinity toward its cognate target, which might appear like noise in the comparison to other high-affinity clonotypes. This diversity in signals is challenging to handle in a one-fit-all filtering process, and for projects with specific interest in low-affinity cell interactions, a specific focus should be addressed not to lose such information.

Effective pairing of TCR and pMHC will open new avenues to interrogate T cell recognition and the role of different T cell populations in pathogenic processes. Intensive efforts have been made to identify antigen specificity based on the TCR sequence (*Gielis et al., 2019*; *Montemurro et al., 2021*; *Moris et al., 2021*; *Sidhom et al., 2021*; *Weber et al., 2021*; *Zhang et al., 2021*), and access to both TCR α- and β-chain is important to improve such prediction strategies (*Montemurro et al., 2021*).

The coveted data is ensured via the ITRAP framework for single-cell data of TCRs and associated barcodes. The perspectives of further exploiting the transcriptomic information, allowing in-depth tracking of specific T cell subsets based on the clonotypes, suggest that we are on the verge of achieving substantial novel insight to T cell involvement and behavior in health and disease.

In conclusion, we have demonstrated that ITRAP is a highly flexible tool for denoising sc-pMHC TCR data to identify the most likely pMHC-TCR pairs. The validation of the tool will benefit from future studies where the pMHC specificities of TCRs may be experimentally validated. Here, we provide a first important step toward this by enabling researchers identifying the most likely pair to proceed with also for very low frequent T cell populations.

## Methods

All healthy donor material was collected from the central blood bank at the university hospital, Rigshospitalet, Copenhagen (under the agreement no. BC29). Collection was done under approval by the Scientific Ethics Committee of the Capital Region, Denmark, and written informed consent was obtained according to the Declaration of Helsinki. This material is fully anonymized.

### Cell samples

PBMCs from healthy donors were isolated from whole blood by density centrifugation on Lymphoprep (Axis-Shield PoC) and cryopreserved at −150°C in FCS (Gibco) + 10% DMSO.

### DNA barcodes and dextran conjugation

Oligonucleotides modified with a 5′ biotin tag were purchased from LCG Biosearch Technologies (Denmark). Read from 5′ to 3′, the oligonucleotides were designed with the 10x equivalent Read2N sequence, a 10 nt UMI, a distinct 15mer nucleotide sequences (extracted from *Xu et al., 2019*), a 9 nt UMI and ending in a 13 nt capture sequence complementary to the TSO of the 10x 5′ capture oligo (sequences are listed in *Supplementary file 1*). Barcodes were dissolved to 100 μM in RNAse-free water and stored at −20°C. For a working solution, the DNA barcodes were further diluted in PBS + 0.5% BSA + 1 mg/mL herring DNA + 2 mM EDTA to 2.17 μM and attached to PE- or APC- and streptavidin-conjugated dextran from FINA Biosolutions LCC (USA). The amount of DNA barcode attached to each new lot of dextran was titrated as described in *Bentzen et al., 2016*. DNA barcodes were attached by mixing with dextran-conjugate, followed by incubation, 30 min at 4°C. DNA barcode-assembled dextran-conjugates were stored for up to 24 hr at 4°C.

## Peptides and MHC monomer production

Peptides were purchased from Pepscan (Pepscan Presto) and dissolved to 10 mM in DMSO. UV-sensitive ligands were synthesized as previously described (*Bakker et al., 2008*; *Rodenko et al., 2006*; *Toebes et al., 2006*). Recombinant HLA-A*0201, HLA-A*0301, and HLA-B*0702, heavy chains, and human β2 microglobulin light chain were produced in *Escherichia coli*. HLA heavy and light chains were refolded with UV-sensitive ligands and purified as described in *Hadrup et al., 2009*. Specific peptide-MHC complexes were generated by UV-mediated peptide MHC exchange (*Chang et al., 2013*; *Frøsig et al., 2015*; *Rodenko et al., 2006*; *Toebes et al., 2006*).

## Generation of DNA barcode-labeled MHC multimer libraries

Unoccupied SA-binding sites on the DNA barcode-assembled dextran conjugates were used for the co-attachment of biotinylated pMHC molecules. 0.8 pmol pMHC monomer was mixed with 160× 10–15 mol DNA-barcoded dextran-conjugate and incubated 30 min at room temperature (RT). MHC multimers were diluted in PBS with 5.2 µM d-biotin (Avidity, Bio200) to 909 nM and incubated 20 min on ice. DNA-barcoded MHC multimers were stored for up 1 wk at −20°C (PBS supplemented with glycerol and BSA, final concentrations 5 and 0.5%, respectively). Immediately before staining barcode-labeled MHC multimers were thawed at 4°C, centrifuged (5 min at 3300 × *g*), and pooled (0.8 pmol of each pMHC/sample) to enable the detection of multiple T-cell responses in parallel. The pooled MHC multimers were centrifuged once more; 5 min at 3300 × *g*, to sediment aggregates before the volume of the reagent pool was reduced by ultrafiltration to obtain a final volume of ~80 µL of MHC multimers as described in *Bentzen et al., 2016*. Any aggregates in the MHC multimer reagent pool were sedimented by centrifugation, 5 min at 3300 × *g* before addition to the cell sample.

## MHC multimer staining

Cryopreserved PBMCs were thawed and washed by sedimentation, 5 min, 390 × *g*, 4°C, in RPMI + 10% FCS. Cells were further washed in a barcode-cytometry buffer (PBS + 0.5% BSA). $5 \times 10^6$ cells were incubated, 60 min, 4°C, with pooled DNA-barcoded multimers in a total volume of 100 µL (final concentration of each distinct pMHC, 8 nM), and washed three times by sedimentation, 5 min, 390 × *g*, 4 °C. 5 µL of Human TruStain FcX Fc Blocking reagent was added to a total of 50 µL cell suspension, and incubated 10 min, 4°C. Hashing antibodies (BioLegend, TotalSeq-C0251 anti-human Hashtag 1-10 Antibodies) were centrifuged 10 min, 14,000 × *g*, 4°C, and 0.5 µL were added to each a distinct sample (*Supplementary file 2*), and incubated 15 min, 4°C. Next a 5× antibody mix composed of CD8-BV480 (BD 566121, clone RPA-T8) (final dilution 1/50), dump channel antibodies: CD4-FITC (BD 345768) (final dilution 1/80), CD14-FITC (BD 345784) (final dilution 1/32), CD19-FITC (BD 345776) (final dilution 1/16), CD40-FITC (Serotech MCA1590F) (final dilution 1/40), CD16-FITC (BD 335035) (final dilution 1/64), and a dead cell marker (LIVE/DEAD Fixable Near-IR; Invitrogen L10119) (final dilution 1/1000), was mixed for each sample. The antibody mix was added to cell samples and incubated 30 min, 4°C. Cells were washed three times in barcode-cytometry buffer and kept on ice until acquisition.

## Cell sorting

Cells were sorted on a FACS Melody (BD) into tubes containing 100 µL of PBS + 0.5% BSA (tubes were saturated with PBS + 2% BSA in advance). Using FACS Chorus software, we gated on single, live, CD8-positive and 'dump' (CD4, 14, 16, 19, and 40)-negative lymphocytes and sorted only APC-positive (PE-negative) cells within this population (*Figure 1—figure supplement 1* for gating strategy). Cells sorted from individual samples were collected into the same tube (*Figure 1b*). The sorted cells were centrifuged for 10 min at 390 × *g,* and the buffer was removed.

## DNA barcode-labeled MHC multimer stained cells on 10x platform

We utilize the 10x 5′ v2 chemistry that allows the cell barcode to be appended at the 5′-end of transcripts, which is essential for capturing the CDR3 region of the V(D)J transcripts. In the 5′ chemistry, the template switch oligo (TSO) is encoded with a cell barcode, that is, one unique 10x barcode for every GEM. The TSO thus comprises the capture oligo, whereas the poly-dT primer is added in free suspension. Reverse transcription is initiated from binding of the poly-dT primer at the 3′-end, and mRNA is captured when the reverse transcriptase enzyme switches at the 5′-end of the transcript

to the TSO. All DNA barcodes, partially complementary to the 10x Genomics 5′ TSO, are captured directly onto the GEMs. Annealing and extension during the reverse-transcription reaction associate the cell barcode and UMI from the gel-bead oligo with the pMHC and hashing antibody tags in parallel with the mRNAs in the same droplet.

Downstream processing of mRNA and DNA barcodes is performed according to the manufacturer's instructions (Chromium Next GEM Single Cell 5' Reagent Kits v2 [Dual Index], with the Feature Barcode technology for Cell Surface Protein & Immune Receptor Mapping) (10x Genomics, USA). Approximately 15,700 cells were loaded (based on 55% recovery from 28,800 sorted cells) to yield a maximum of 9000 cells with an intermediate/high doublet rate (6,9%). Targeted amplification was performed for 13 cycles and the products were separated according to size into <400 bp (DNA barcode-tags) and >400 bp (the TCR sequences) using 0.6× SPRIselect beads (Beckman Coulter, B23318). Separate processing of the >400 bp bead-bound TCR sequences and the <400 bp in solution DNA barcodes was conducted according to the manufacturer's instruction and the TCR amplification products were sequenced on a NovaSeq running a 150 paired-end program. DNA barcodes, TCR sequences, and mRNA were sequenced to a depth of 13,332, 12,503, and 18,398 mean reads per cell, respectively.

## Bioinformatics

### Processing of 10x single-cell data

Hashing barcode reads, peptide-MHC barcode reads, and T cell gene expression reads were provided in fastq format and were processed using 10x Genomics Cellranger multi v6.1.0 (*10xGenomics, 2022b*). The relevant outputs were the unfiltered count matrices of DNA barcodes and gene expression as well as clonotype annotations of each sequencing contig containing CDR3α/β sequences, V(D) J-C genes, and UMI counts.

### Postprocessing 10x Cellranger clonotyping

The raw contig annotations from Cellranger were selected for downstream analysis with filtering on incomplete and unproductive receptor transcripts. Incomplete contigs are not full length, that is, do not span the V-gene start codon until the J-gene stop codon. Unproductive contigs contain a frameshift that either induces an early stop codon or completely removes the stop codon.

Clonotypes defined by 10x were merged when consisting of identical VJ-CDR3αβ, thus reducing functional duplicates.

Cellranger flags rare nucleotide transcripts as likely artifacts, meaning the GEMs are flagged as unlikely to contain a cell and are therefore not assigned a clonotype (*10xGenomics, 2022a*). Therefore, GEMs that were not annotated with a clonotype were imputed by searching the duplicate-reduced clonotype set. If no match, a new clonotype ID was annotated to the GEM.

### Filtering based on gene expression

Filtering on gene expression data was performed as described in *Zhang et al., 2021*. Low-quality GEMs such as doublets may be removed by excluding GEMs with more than 2500 genes. Dead cells may be removed by excluding GEMs with fewer than 200 genes and a ratio of mitochondrial gene expression to the total gene expression above 0.2.

### Demultiplexing samples via cell hashing

Cell Hashing uses oligo-tagged antibodies against ubiquitously expressed surface proteins to place a sample barcode on each single cell, enabling different samples to be multiplexed together and run in a single experiment. To demultiplex the samples, the method presented by Stoeckius et al. was implemented (*Stoeckius et al., 2018*). The method clusters the normalized count matrix using k-medoid clustering into k clusters, $k = n_{samples} + 1$. For each barcode, a negative binomial distribution is fitted to the pool of all clusters except the cluster with the highest average expression for the given barcode. Each GEM is classified as positive if the barcode value exceeds a 0.99 quantile threshold for the negative distribution, and otherwise classified as negative. If GEMs contain multiple barcodes that pass the threshold, the GEM is annotated as a doublet (*Stoeckius et al., 2018*).

## Defining the expected binder

The pMHC and cell hashing barcode annotations were merged with the T cell annotations on the GEMs that contained both TCR and pMHC attributes. Each clonotype is expected to have a preferred target within the pMHC library, thus each clonotype is evaluated to find the pMHC, which is most likely to be that target.

Each clonotype is evaluated to identify the expected target within the pMHC library. The pMHCs that are detected within the GEMs annotated to a given clonotype are compared by their UMI count distribution. The two pMHCs that have the highest mean UMI count are compared, testing the hypothesis that the expected binder will have a significantly higher mean UMI count than the other pMHC (Wilcoxon, $\alpha$ = 0.05). Clonotypes of less than 10 GEMs were not tested. The clonotypes where the mean UMI of the top two pMHCs was significantly different were collected as a training set. The pMHC that had significantly higher mean UMI was annotated as the expected target and specificity annotations of the GEMs were individually evaluated. True interactions were then defined as GEMs where the most abundant pMHC matched the identified expected target. The GEMs where the most abundant pMHC did not match the expected target were labeled as false interactions. The GEMs belonging to clonotypes where no expected target was identified were left out.

## Defining specificity concordance

Concordance is an indirect measure of how cross-reactive a certain clonotype is. Specificity concordance is defined as the ratio of GEMs of a single clonotype that are annotated to bind a particular pMHC. The more GEMs in a clonotype annotated to the same pMHC the larger concordance. If a clonotype is only detected with one pMHC, the specificity concordance is 1.

## Grid search on UMI features

Based on the labels of the training set, a performance metric, o, was defined to evaluate the accuracy at increasing thresholds for UMI count and UMI ratio of pMHC, α-chain, and β-chain. The UMI ratio measures multiplets and is defined as the ratio between the highest UMI count and the second highest UMI count in a GEM:

$$UMI_{ratio} = \frac{UMI_{max}}{UMI_{sec} + 0.25}$$

A small number (0.25) was added in the denominator to avoid division by zero.

The performance metric, o, is a weighted average of accuracy and fraction of retained GEMs, given by the following equation:

$$o = \frac{2 \cdot acc + f_{retained\ GEMs}}{3}$$

The accuracy metric is defined by the ratio of training set GEMs that were labeled as true interactions over the total number of GEMs in the training set. The performance metric, o, was maximized by finding the set of filters that increase the accuracy without excluding too much data.

The thresholds for filtering were selected from a complete grid search. Each feature was tested in the range of 0 to the median value, determined ad hoc from the experience that thresholds never approached the median value.

## Comparing TCR similarity

Effects of filtering were also evaluated through a comparison of TCR similarity. The similarity score is based on the kernel similarity score underlying the TCRmatch method between CDR3 sequences (**Chronister et al., 2021**). This score can be calculated for CDR3s of variable length and takes a value between 0–1, with the value of 1 representing identical pairs. As both the α- and β-chain partake in the pMHC interaction, TCRs will be compared based on the summed similarity between the α- and β-chains, and GEMs missing a chain will be excluded to avoid bias. Two similarity scores are computed for each unique TCRαβ-pair (per clonotype): an intra-score and an inter-score. The intra-score is based on the maximum similarity of the given clonotype to all other clonotypes sharing its pMHC target (intra-specificity). The inter-score is based on the maximum similarity of the given clonotype to an

equal-sized set of clonotypes specific to other pMHC targets (inter-specificity). The computation is done peptide-wise, such that clonotypes with maximum concordance for a given peptide will, for that peptide, be included in an intra-similarity score, but for another peptide be included in an inter-similarity score. Clonotypes consisting of GEMs causing diverging specificities were limited to the expected target pMHC or, if nonexisting, to the specificity of highest concordance to avoid potential overlaps from 'cross-reactive' clonotypes in the computation.

The similarity difference between intra- and inter-specificity clonotypes was tested for the hypothesis that intra-similarity is greater than inter similarity (Wilcoxon, $\alpha = 0.05$). The similarity test was performed on all filtering methods described in the paper.

## Validating single-cell capture against fluorescent multimer staining responses

The 16 donors were known to respond to the panel of peptides used in the screening. Response proportions of sorted CD8+ T cells were detected by fluorescent multimer staining, as described previously. A total of 1800 cells were selected from each donor and, based on the detected response proportions, an adjusted count of cells could be computed. Cells were selected based on two criteria: ≥10 cells and ≥0.002% of total CD8 T cells, or ≤10 cells but ≥0.01% of total CD8 T cells. The multimer responses were compared to GEMs filtered on UMI thresholds and matching HLA. To visually compare the two screening methods, the responses were normalized within each sample and plotted side-by-side. The methods were also quantitatively compared, both in absolute counts of responses and as binary classes with multimer responses as true labels and single-cell responses as query labels.

The correlation in *Figure 7d* was also fitted via linear regression on log10 transformed data, resulting in the following equation:

$$log_{10}\left(y\right) = 0.86 \cdot log_{10}\left(x\right) + 1.18, \ R^2 = 0.56$$

The equation was used to estimate the yield of single-cell captured cells relative to multimer screening. Three examples were computed to estimate an approximate 10% yield.

$$log_{10}\left(10\right) = 0.86 \cdot log_{10}\left(0.6\right) + 1.18$$
$$log_{10}\left(100\right) = 0.86 \cdot log_{10}\left(9.0\right) + 1.18$$
$$log_{10}\left(1000\right) = 0.86 \cdot log_{10}\left(131.2\right) + 1.18$$

# Acknowledgements

We would like to thank all healthy donors contributing material to this study. This research was funded in part through the Independent Research Fund Denmark (DFF 7014-00055 to SRH and MN), the Lundbeck Foundation (R322-2019-2445 and R324-2019-1671 to AKB and R190-2014-4178 to SRH), the European Research Council, StG 677268 NextDART to SRH, the European Union's Horizon 2020 research and innovation program under the Marie Sklodowska-Curie grant agreement no. 713683 (COFUNDfellowsDTU), and National Institute of Allergy and Infectious Diseases (NIAID), under award number 75N93019C00001 to HRP and MN.

# Additional information

### Competing interests
Amalie Kai Bentzen, Sine Reker Hadrup: AKB and SRH are co-inventors on a patent covering the use of DNA barcode labeled MHC multimers (WO2015185067 and WO2015188839), which is licensed to Immudex. The other authors declare that no competing interests exist.

## Funding

| Funder | Grant reference number | Author |
| --- | --- | --- |
| Danmarks Frie Forskningsfond | 7014-00055 | Sine Reker Hadrup |
| Lundbeckfonden | R322-2019-2445 | Amalie Kai Bentzen |
| Lundbeckfonden | R324-2019-1671 | Amalie Kai Bentzen |
| Lundbeckfonden | R190-2014-4178 | Sine Reker Hadrup |
| European Research Council | StG 677268 NextDART | Sine Reker Hadrup |
| HORIZON EUROPE Marie Sklodowska-Curie Actions | 713683 | Helle Rus Povlsen |
| National Institute of Allergy and Infectious Diseases | 75N93019C00001 | Helle Rus Povlsen |

The funders had no role in study design, data collection and interpretation, or the decision to submit the work for publication.

## Author contributions

Helle Rus Povlsen, Conceptualization, Data curation, Software, Formal analysis, Validation, Investigation, Visualization, Methodology, Writing – original draft, Writing – review and editing; Amalie Kai Bentzen, Conceptualization, Data curation, Formal analysis, Validation, Investigation, Methodology, Writing – original draft, Writing – review and editing; Mohammad Kadivar, Conceptualization, Writing – review and editing; Leon Eyrich Jessen, Supervision, Writing – original draft, Writing – review and editing; Sine Reker Hadrup, Conceptualization, Resources, Supervision, Funding acquisition, Methodology, Project administration, Writing – review and editing; Morten Nielsen, Resources, Software, Formal analysis, Supervision, Funding acquisition, Writing – original draft, Project administration, Writing – review and editing

## Author ORCIDs

Helle Rus Povlsen http://orcid.org/0000-0002-5687-4386
Mohammad Kadivar http://orcid.org/0000-0003-1499-191X
Leon Eyrich Jessen http://orcid.org/0000-0003-2879-2559

## Ethics

Human subjects: All healthy donor material was collected from the central blood bank at the university hospital, Rigshospitalet, Copenhagen (under the agreement no. BC29). Collection was done under approval by the Scientific Ethics Committee of the Capital Region, Denmark, and written informed consent was obtained according to the Declaration of Helsinki. This material is fully anonymized.

## Decision letter and Author response

Decision letter https://doi.org/10.7554/eLife.81810.sa1
Author response https://doi.org/10.7554/eLife.81810.sa2

# Additional files

## Supplementary files

• Supplementary file 1. Samples. Overview of which samples contain cells from which donors and the relevant donor haplotypes.

• Supplementary file 2. Peptide-MHC multimers. Information on the applied pMHC multimers. The full oligonucleotide tag are designed as follows: Biotin-C6-CGGAGATGTGTATAAGAGACAGNN NNNNNNNNXXXXXXXXXXXXXXXXXXXNNNNNNNNNNCCCATATAAGAAA, with the barcode sequence indicated by 15 purple X's. C6 indicates a six carbon spacer with a hydroxyl to the 5' end of an oligonucleotide. Read2N is indicated by the black sequence, UMIs are indicated in gray, and the capture oligo is indicated in turquoise.

• Supplementary file 3. Database cross-referencing specificities. Information on the CDR3 sequences that matched the CDR3 sequences of the IEDB and VDJ databases presented in *Figure 2d*. Six

different clonotypes (ct) had CDR3 sequence matches. Five of the clonotypes also matched (T:True) the database on the annotated pMHC (DB Match), while one clonotype (ct 573) had conflicting annotations.

• Supplementary file 4. Multimer staining responses. The frequency or summed number of gel-beads in emulsion (GEMs) for all pMHCs reported in *Figure 7*. The frequency found by fluorescent-based methods is utilized to calculate a proportion of each response per donor ((sum of responses in donor/100) × % of pMHC-specific T cells) and then the estimated number of cells of each specificity sorted for the single-cell analyses ((total number of cells sorted × proportion)/100). In all cases, a total of 1800 cells were sorted. Approximately 45% of sorted cells are lost already before loading on 10x, and 50% more are expected to be lost during 10x processing. Responses reported in italic are below the detection limit for the fluorescent-based method but represent cases where >1 event are detected for that specificity. NA, not applicable.

• Supplementary file 5. Singlet summary of *Figure 6c*. Summary of specificity singlets, that is, specificities only supported by one gel-bead in emulsion (GEM). The three columns of singlet frequencies correspond to the data plotted in *Figure 6a–c* which represent: (a) raw data, (b) data filtered on unique molecular identifier (UMI) thresholds and HLA match, and (c) data filtered on UMI thresholds, HLA match, and only contains complete TCRs (TCRαβ). A singlet frequency of 1.0 corresponds to 100% specificity singlets.

• MDAR checklist

## Data availability

The different data sets analyzed and generated in this study are available at https://doi.org/10.11583/DTU.22645342.v1. These data includes the raw data file (raw.csv), the data filtered by the optimized UMI count thresholds (opt_thr.csv), the data filtered by the UMI thresholds and HLA matching (hla_match.csv), and final filtered data including only GEMs with complete TCR annotation. The ITRAP code for optimal UMI count threshold identification, and subsequent filtering is available at https://github.com/mnielLab/itrap (copy archived at *Povlsen and Nielsen, 2023*).

The following dataset was generated:

| Author(s) | Year | Dataset title | Dataset URL | Database and Identifier |
|---|---|---|---|---|
| Nielsen M, Povlsen HR, Bentzen AK, Kadivar M, Jessen LE, Hadrup SR | 2023 | Data for: Accurate T cell Receptor Antigen Pairing through data-driven filtering of sequencing information from single-cells | https://doi.org/10.11583/DTU.22645342.v1 | Technical University of Denmark, 10.11583/DTU.22645342.v1 |

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
