## [Editor Report]

This paper is of interest to immunologists conducting single-cell analyses of T-cell recognition. It provides improved means of curating datasets to reduce noise and identify T cell-antigen pairs with greater confidence. Experimental data from human virus-specific TCRs are used to validate the methodology.

---

## [Decision Letter]

**Decision letter after peer review:**

Thank you for submitting your article "ATRAP – Accurate T cell Receptor Antigen Pairing through data–driven filtering of sequencing information from single–cells" for consideration by *eLife*. Your article has been reviewed by 3 peer reviewers, and the evaluation has been overseen by a Reviewing Editor and Tadatsugu Taniguchi as the Senior Editor. The following individual involved in the review of your submission has agreed to reveal their identity: Michael E Birnbaum (Reviewer #1).

This paper is of interest to immunologists conducting single–cell analyses of T–cell recognition. It provides a means of curating datasets to ensure T cell–antigen pairs are identified. However, the data generated through this method often suffers from a relatively high background. The authors present a computational approach to enhance signal–to–noise of this type of analysis. However, from the reviewer's comments, it is unclear if the thresholds and filtering steps described by the authors can be generally applied to other datasets of different qualities than the one used here. The reviewers make suggestions to better stress–test the robustness of the method. Overall this is a potentially valuable contribution but requires additional benchmarking and more clarity on the limitations of the approach.

Essential revisions:

1) The manuscript is more suitable for an *eLife* Resource given that it is entirely methodological and does not shed new biological insight.

2) There is a need for benchmarking the data against other datasets to assess the robustness and optimal threshold selection.

3) Ensure that the code and data used in the manuscript are publicly available so others can use the method.

4) Address reviewer comments about other technical concerns.

5) Consider the limitations of the study in the revised manuscript and title.

*Reviewer #1 (Recommendations for the authors):*

1. It will be helpful for the figures in the manuscript to be of higher quality (vectorized) for publication – there are areas where figure quality made data interpretation difficult.

2. Line 34 – MHC should be defined.

3. Line 55 – a and b should be changed to α and β.

4. In figure 2, it is difficult to distinguish between HLA match True and False. May it be worth having the two in different colors?

5. Lines 116 – 121: It would help to clarify the use of the PE–multimers a bit more, than by using them while sorting for APC–only, it ensures that any PE–based signal is purely due to solution–based noise rather than cellular contamination. This was something I only fully understood after reading through the paper once.

6. In lines 160–168, the authors state that 28000 cells were sorted and 45% of the cells were lost in the loading process. It would be helpful if the authors could clarify how were these numbers generated. Were 28000 cells a proxy for 28000 sorted events? How did the authors know that 45% of cells were lost in the process and only 15,700 cells were loaded on the 10X?

7. Based on the observations and discussion in lines 288 through 298, it might be helpful if the authors explicitly stated that they are defining true binders = expected binder=significantly most abundant pMHC for a given clonotype, rather than as defined through orthogonal means.

8. On line 204 the authors mention that the three negative control pMHCs were only present in a few GEMs. However, are these barcodes were captured as ambient contamination or were they captured with distinct clonotypes?

9. Line 274–275: The authors state that for GEMs with TCR multiplets, they take the most abundant chain for analyses. It would help if this were clarified. Is this the most abundant UMI per GEM (so if a given GEM has ⍺1 with 10 UMIs and ⍺2 with 8 UMIs, it counts as ⍺1 ), or the most abundant call per clonotype? (so if there are 5 GEMs with ⍺1 with the most UMIs and 3 GEMs with ⍺2 with the most UMIs, the entire clonotype is called for ⍺1 )? Does this analysis already remove any ⍺ transcripts that are recombined but out of frame or contain stop codons?

10. Line 479–482: The percentages for HLA match and mismatch are provided – what would the probabilities be expected if by chance?

11. Lines 531–535: The authors should further clarify why differences in fluorescence signal may account for differences in analysis for the single–cell sequencing vs FACS, especially given the fact that the sequenced cells are also sorted via FACS before 10x analysis. Is there a difference in avidity and/or concentration between the two staining reagents used?

*Reviewer #2 (Recommendations for the authors):*

1) Please explain what was the motivation for doing the experiment. Are the donors seropositive/seronegative for CMV/EBV/Flu? Why were these particular epitopes selected? What was the phenotype of sorted cells? What was the hypothesis?

2) Please make the raw data, processed data, and code available. The main strength of the paper is the robust code which could be potentially used to clean up other datasets of the same kind. The link to github returns 404 error.

3) It seems that the approach is not robust in the presence of cross–reactivity. If there are two different pMHC complexes loaded with highly similar peptides recognized by the same clone (and thus two pMHCs with different barcodes bound to the same cell), how the specificity will be assigned (and how will it influence UMI threshold selection, lines 344–345)?

4) It seems that TCR similarity metrics (both for inter and intra–similarity) are defined as maximal similarity values across all the comparisons within the same peptide assignment, or others, lines 888–891. This value should be systematically biased by the sample size (the more pairwise comparisons we do, the more extreme similarity we will find, even if the underlying sequence distance distribution is the same). It is not clear to me, how authors normalize this effect (do they downsample to the minimal number of unique clonotypes across all epitopes)?

5) Figure S1 shows sharing of VJCDRab between GEMs. How will this plot look if we consider sharing of a single chain nucleotide sequence (VJCDR3a_nucleotide or VJCDR3bnucleotide) between GEMs? If a clone has two alphas, will the proposed pipeline split it into two different clonotypes?

6) Please discuss and compare the data analysis strategy to the one from the following recent manuscripts:

https://www.science.org/doi/10.1126/sciimmunol.abk3070

https://www.nature.com/articles/s41590–022–01184–4

https://www.ncbi.nlm.nih.gov/pmc/articles/PMC9184244/

7) Line 318 mentions Figure 1d (probably instead of 2d)

*Reviewer #3 (Recommendations for the authors):*

– It will be clearer if the authors could provide a workflow of ATRAP that describes the key steps in the bioinformatic process.

– The cell number loaded on the 10X Chromium for each donor shown in supplementary table 4 indicated the GEM counts for each donor were less than 264. It would be important for the authors to comment on cell numbers and what threshold would be sufficient to perform this method. This would enable future users of the technique to have more guidance in decision–making.

– For experimental control, the authors only use three additional pMHC multimers bearing a different fluorochrome (PE). However, it would be more interesting to see if the authors could include negative pMHC multimers bearing the same fluorochrome (APC) to estimate the background binding noise for each donor and to check if the ATRAP could successfully remove those negative pMHC multimers.

– In this study, the author only used IEDB (Vita et al., 2019) and VDJ (Bagaev et al., 2020) databases to prove the detected specificities in their data have been cross–referenced on only five clonotypes. However, the author did not provide experimental evidence showing the pMHC selected by the ATRAP is a real target of a specific clonotype and that those pMHCs removed by the ATRAP are not the target of that clonotype. This may be a rather intensive set of experiments to show this, but the authors could consider it or at least make some statements/caveats if they choose not to do such additional validation.

– At lines 654–659, the authors write that low–avidity clonotypes might appear like noise, but this method is only able to detect the binding affinity, not the avidity. The binding affinity of TCR is not always correlated with avidity. I wonder if the information in their dataset really provides avidity measurements.

– At lines 344–345, it said that "This filtering analysis resulted in optimal thresholds of 2 pMHC UMI counts and a ratio pMHC UMI counts between top one and two >1". Is it possible if the sequence results are deeper that it might result in more noise or background pMHC UMI counts, in such a case, how would one adjust the optimal thresholds?

– In general, I think readers would find the V and J gene usage, along with other immune repertoire information interesting for all pMHC binders. The authors should consider this, perhaps as supplementary data.

---

## [Author Response]

Essential revisions:1) The manuscript is more suitable for an eLife Resource given that it is entirely methodological and does not shed new biological insight.

We have no objections to this if the editor agrees.

2) There is a need for benchmarking the data against other datasets to assess the robustness and optimal threshold selection.

We have described such benchmark comparisons in a separate manuscript submitted for publication and currently available in BioRxiv doi.org/10.1101/2023.02.01.526310. Here ITRAP was compared to ICON the, to the best of our knowledge, only other currently available method for denoising of single cell obtained TCR-pMHC data on the large set of data generated from 10X Genomics. We have added a short section to the manuscript Discussion section describing the outcome of this benchmark and how the results compare to the result described in the current manuscript.

3) Ensure that the code and data used in the manuscript are publicly available so others can use the method.

We have made all data from the manuscript publicly available at https://services.healthtech.dtu.dk/suppl/immunology/ITRAP. Further, we have made the code for the ITRAP threshold filtering step available for download at https://services.healthtech.dtu.dk/suppl/immunology/ITRAP/.

4) Address reviewer comments about other technical concerns.

We believe to have carefully addressed all the reviewers comments related technical concerns in the responses below.

5) Consider the limitations of the study in the revised manuscript and title.

We have discussed in detail the limitations of the study in the current manuscript, and have updated the title and method acronym to ITRAP (Improved T cell Receptor Antigen Pairing) replacing the word “accurate”. We are naturally very happy that *eLife* finds our work of interest. Regarding the specific question related to if the thresholds and filtering steps described in the current can be applied to other data sets, we have, as described above, conducted an analysis on the 10X Genomics data set in a separate manuscript. Here, we demonstrate that the method is both generally applicable to any single cell data set with UMI count information related to pMHC and TCR. We also demonstrated how the denoising can be highly improved if information about donor origin is included in terms of cell-hashing. Further, we find that the impact of the different filtering steps contained in ITRAP aligns well between the two studies. In one aspect the results however are somewhat different. This is with respect to the actual thresholds identified for the UMI filtering. Here, the values from the 10X data as expected were different compared to the values found in the current manuscript (see Author response table 1). These differences however in our view indicate the high strength of ITRAP being robust at being applied to data generated in very different biological and technical settings. We have extended the manuscript with a section on this independent data set analysis and a discussion of the results.

**Author response table 1. sa2table1:** Overview of UMI thresholds given by ITRAP. The table provides the UMI thresholds for each component (pMHC, TRA, and TRB) identified using the accuracy optimizing approach of ITRAP. Beyond direct UMI thresholds, data may also be filtered based on the UMI ratio between the two largest UMI measurements within a GEM (per component). Two sets of thresholds are given, as the thresholds are dependent on the dataset: first set is based on the data of this publication, the second set is based on the publicly available data by 10x Genomics presented in Povlsen & Montemurro, 2023 (DOI 10.1101/2023.02.01.526310).

Variable	Current manuscript	10x benchmark
UMI pMHC	2.0	5.0000
UMI ratio pMHC	1.0	1.1684
UMI TRA	0.0	1.0000
UMI ratio TRA	0.0	0.7579
UMI TRB	0.0	0.0
UMI ratio TRB	0.0	0.7579

with “improved”.

Reviewer #1 (Recommendations for the authors):1. It will be helpful for the figures in the manuscript to be of higher quality (vectorized) for publication – there are areas where figure quality made data interpretation difficult.

We are sorry about this, and have updated the figure quality in the revised manuscript.

2. Line 34 – MHC should be defined.

Resolved.

3. Line 55 – a and b should be changed to α and β.

Resolved.

4. In figure 2, it is difficult to distinguish between HLA match True and False. May it be worth having the two in different colors?

Resolved.

5. Lines 116 – 121: It would help to clarify the use of the PE–multimers a bit more, than by using them while sorting for APC–only, it ensures that any PE–based signal is purely due to solution–based noise rather than cellular contamination. This was something I only fully understood after reading through the paper once.

We understand why this can be confusing. We have updated the text to give a more thorough description for the motivation behind the experimental design, which should make the use of the PE labeled multimers more clear.

6. In lines 160–168, the authors state that 28000 cells were sorted and 45% of the cells were lost in the loading process. It would be helpful if the authors could clarify how were these numbers generated. Were 28000 cells a proxy for 28000 sorted events? How did the authors know that 45% of cells were lost in the process and only 15,700 cells were loaded on the 10X?

We have updated the text to make it evident that this number of sorted cells is indeed a proxy for an appropriate number of cells to be loaded on the 10x Chromium, taking into account our specific experiences with the number of cells lost in and immediately after the sorting process.

7. Based on the observations and discussion in lines 288 through 298, it might be helpful if the authors explicitly stated that they are defining true binders = expected binder=significantly most abundant pMHC for a given clonotype, rather than as defined through orthogonal means.

We agree and have updated the wording in line 313 to:

“Having annotated the “expected” pMHC of a given clonotype”.

in the revised manuscript. We believe this also help clarify the subsequent text in line 314:

“where the most abundant pMHC corresponds to the expected binder, as “true”, and all others as “false”, and use these annotations to quantify the accuracy of the GEM annotations”.

8. On line 204 the authors mention that the three negative control pMHCs were only present in a few GEMs. However, are these barcodes were captured as ambient contamination or were they captured with distinct clonotypes?

Note first that the “negative control” pMHC are not truly negative. They are highly dominant pMHC specificities deselected in the sorting prior to the 10x single cell sequencing. It is hence expected that a small proportion of true positive TCRs towards these pMHCs can be found in the data. This is indeed the case, as 4 of the identified TCRs against the control peptides had identical matches within the IEDB/VDJdb database. As stated in the text, the three negative controls were only captured in a few GEMs. We have added the actual numbers (GIL:4 GEMs/Clonotypes, GLC: 17 GEMs/Clonotypes, and NLV: 12 GEMs/Clonotypes) to the manuscript to better underline this. Note, further from these numbers, that all negative control clonotypes are singletons, i.e one GEM per clonotype. -We also observe 107 cases where these pMHCs appear like contamination (i.e. with an UMI count lower than the top one pMHC). In 91 (85%) of these GEMs the negative control was present with 1 UMI, while the UMI counts range from 2-20 in the remaining GEMs. In comparison, a similar proportion of ambient contamination was observed for the CLG peptides. This peptide was found as a “contaminant” in 618 GEMs of which 523 (85%) had an UMI count of 1. These observations strongly suggest that the background of control pMHC barcodes is due to ambient contamination.

9. Line 274–275: The authors state that for GEMs with TCR multiplets, they take the most abundant chain for analyses. It would help if this were clarified. Is this the most abundant UMI per GEM (so if a given GEM has ⍺1 with 10 UMIs and ⍺2 with 8 UMIs, it counts as ⍺1 ), or the most abundant call per clonotype? (so if there are 5 GEMs with ⍺1 with the most UMIs and 3 GEMs with ⍺2 with the most UMIs, the entire clonotype is called for ⍺1 )? Does this analysis already remove any ⍺ transcripts that are recombined but out of frame or contain stop codons?

This is a detailed observation, and a very good question. We assume the reviewer is referring to the text in lines 269-270 of the original submission. Clonotypes are defined from the TCRs present in the GEMs. This means that individual GEMs within a given clonotype can be TCR multiplets and have ambiguous TCR associations. The ITRAP pipeline does not resolve such ambiguities. This is done by the user and any given point in the filtering pipeline. In our case, this was done prior to conducting the similarity analysis. Here TCR annotations for individual GEMs were generated from the highest UMI count values. We have updated the text to clarify this in the revised manuscript.

10. Line 479–482: The percentages for HLA match and mismatch are provided – what would the probabilities be expected if by chance?

This is a valid question. Here rather than addressing the random distribution, we have calculated the pMHC distribution across the 2777 non-outlier GEMs. Here we find RVR A0301 and TPR B0702 to be present in 34% and 21% of the data, respectively. We have updated the text to include and comment on this information.

11. Lines 531–535: The authors should further clarify why differences in fluorescence signal may account for differences in analysis for the single–cell sequencing vs FACS, especially given the fact that the sequenced cells are also sorted via FACS before 10x analysis. Is there a difference in avidity and/or concentration between the two staining reagents used?

We appreciate the reviewers comment on this important topic.

For the T cell sorting prior to the single-cell analyses we apply dextran-conjugated multimers while for the fluorescent-labeled combinatorial-encoded we applied tetramers. We have previous compared detection using these different types of pMHC reagents, and did not observe any differences for T cells recognizing viral epitopes (Bentzen, Nat Biot 2016). However, the two measures were runned on two different flow cytometers, and used different fluorophores, which can impact the fluorescent separation and cell identification. Furthermore, a threshold of 10 events was applied to the T cell detections strategy using the combinatorial-encoded tetramers, while in the single cell platform, all GEM that passed our filtering process was included. This discrepancy has been clarified in the manuscript (lines 564-567).

Moreover, we would here again like to refer to figure 7, where we demonstrate a very high concordance between the two approaches, reflected by the vast majority of the responses detected by fluorescent multimer staining also being captured in the single-cell screening, (recall of 0.95).

Reviewer #2 (Recommendations for the authors):1) Please explain what was the motivation for doing the experiment. Are the donors seropositive/seronegative for CMV/EBV/Flu? Why were these particular epitopes selected? What was the phenotype of sorted cells? What was the hypothesis?

Please refer to the text above related to the selection of donors and epitopes. It was not essential for us to know the phenotype of the cells in this study, hence we have not investigated this parameter. However, this could be explored looking more into the transcriptomic data which is available and in future experiments by including TotalSeq antibodies for simultaneous staining of surface markers.

2) Please make the raw data, processed data, and code available. The main strength of the paper is the robust code which could be potentially used to clean up other datasets of the same kind. The link to github returns 404 error.

All data and code has been made available at https://services.healthtech.dtu.dk/suppl/immunology/ITRAP

3) It seems that the approach is not robust in the presence of cross–reactivity. If there are two different pMHC complexes loaded with highly similar peptides recognized by the same clone (and thus two pMHCs with different barcodes bound to the same cell), how the specificity will be assigned (and how will it influence UMI threshold selection, lines 344–345)?

This is a very important question. As described above, ITRAP does not exclude cross-reactivity. The filtering thresholds are defined from a subset of clonetypes with well-defined single specificities. While one can argue that a minor limitation of ITRAP is that such well-defined single specificity clonotypes are needed, we believe this will most often be the case. And if not, a limited set of prevalent pMHC targets can be included to ensure this. Once these thresholds are defined, clonotypes with multiple specificities can trivially be identified and analyzed. Note that this is in contrast to what can be archived with for instance the ICON method.

4) It seems that TCR similarity metrics (both for inter and intra–similarity) are defined as maximal similarity values across all the comparisons within the same peptide assignment, or others, lines 888–891. This value should be systematically biased by the sample size (the more pairwise comparisons we do, the more extreme similarity we will find, even if the underlying sequence distance distribution is the same). It is not clear to me, how authors normalize this effect (do they downsample to the minimal number of unique clonotypes across all epitopes)?

This is an important and insightful observation. In the inter versus intra similarity analysis, all clonotypes within a given plateau (with a given annotated peptide specificity) are compared against all other clonotypes within the plateau and against the same number of clonotypes randomly sampled from all other plateaus. It is clear that this sampling approach could result in biases associated with the size of the given plateau. It is however not to us trivially clear whether this bias would be in favor of larger or short plateaus having a significant difference between inter and intra similarity scores. It is trivially clear that the intra similarity score will be biased towards higher values for larger plateaus, but the same is the case for the inter similarity score (since we here also are comparing against a larger set of TCRs from other plateaus). Given this we believe our current simple approach is fair and suitable for capturing the proposed signal.

5) Figure S1 shows sharing of VJCDRab between GEMs. How will this plot look if we consider sharing of a single chain nucleotide sequence (VJCDR3a_nucleotide or VJCDR3bnucleotide) between GEMs? If a clone has two alphas, will the proposed pipeline split it into two different clonotypes?

The pipeline does as such not refine clonotypes. Clonotypes are defined from the TCR annotations of 10X CellRanger. The only modifications are the cases listed in the section “Clonotype annotation” and covers “we redefine clonotypes based on TCR annotation. Subsequently, GEMs with no clonotype annotation from 10x were annotated to existing clonotypes conditioned on matching VJαβ-genes and CDR3αβ sequences or as novel clonotypes. Similarly, clonotypes with identical VJ-CDR3αβ were merged to form larger groups of theoretically identical TCRs”.

While we agree that a further analysis of the sharing of VJCDR between GEMs is interesting, we believe that such an additional analysis split on VJCDR3a_nucleotide or VJCDR3b_nucleotide is beyond the scope of the current paper.

6) Please discuss and compare the data analysis strategy to the one from the following recent manuscripts:https://www.science.org/doi/10.1126/sciimmunol.abk3070https://www.nature.com/articles/s41590–022–01184–4https://www.ncbi.nlm.nih.gov/pmc/articles/PMC9184244/

All these studies are related to data generation rather than data denoising. As stated above, we are currently in the process of analyzing these data using the ITRAP framework. We however believe these analyses are beyond the scope of the current publication. Note also that we, as stated above, have conducted a benchmark experiment of the data set provided by 10x Genomics. The results is this study described in an separate publication [BioRxiv doi.org/10.1101/2023.02.01.526310]

7) Line 318 mentions Figure 1d (probably instead of 2d)

Resolved.

Reviewer #3 (Recommendations for the authors):– It will be clearer if the authors could provide a workflow of ATRAP that describes the key steps in the bioinformatic process.

We thank the reviewer for this suggestion and have added a workflow of ITRAP to the revised manuscript (new Supplementary figure 6).

– The cell number loaded on the 10X Chromium for each donor shown in supplementary table 4 indicated the GEM counts for each donor were less than 264. It would be important for the authors to comment on cell numbers and what threshold would be sufficient to perform this method. This would enable future users of the technique to have more guidance in decision–making.

The number of GEMs captured in a single 10X run is expected to be of the order 8000-10000. Of these only a proportion will end up containing signals for both pMHC and TCR (in our case ~6000). Upon filtering this number is further reduced to ~4000. Splitting these responses among the individual donors (hashing) results in ~200 GEMs per donor (note that hashing entry 10 covers multiple donors, and hence take up a higher proportion of the GEMS). The observed GEM count of 264 or less per donor is hence what should be expected when conducting an experiment with a complexity like the one performed here. Note however, that even with these low GEMs counts per donor, we were able to detect responses that strongly align both qualitatively and quantitatively with what has been observed in the same donors using conventional fluorescent multimer staining. We have elaborated briefly on this in the text, by indicating that the number of captured GEMs with a given TCR-pMHC pair is directly proportional with the input. Hence low numbers of cells included from one sample will result in lower numbers of GEMs for any given pMHC-TCR pair.

We have updated the manuscript to include a discussion of this (lines 579-585).

– For experimental control, the authors only use three additional pMHC multimers bearing a different fluorochrome (PE). However, it would be more interesting to see if the authors could include negative pMHC multimers bearing the same fluorochrome (APC) to estimate the background binding noise for each donor and to check if the ATRAP could successfully remove those negative pMHC multimers.

The PE labeled pMHC multimers were only included as internal controls for us to get a first qualitative estimation of the specificity of the 10X technology. Also refer to previous response for the rationale for including the PE labeled pMHCs, i.e. to actively deselect certain specificities.

We actively chose not to be dependent on experimental controls in the ITRAP method. We believe this is an advantage of the ITRAP approach over for instance ICON where negative controls are explicitly needed in order to define background noise. This requirement imposes an additional cost for barcode MHC-multimer reagents. We agree with the reviewer that it can be valuable for users to investigate their data broadly to “estimate” background signals. However, in most experimental setups where several samples are screened in parallel (including the one included in this manuscript) indirect negative controls will be included. This includes HLA mismatches for the samples, and the notion that no sample is expected to recognise all pMHCs (based on previous screens). Our experience is that these might be more relevant as negative controls, since we can in fact not be sure that a given epitope is not recognised due to cross-recognition events or binding of T cells from the naive repertoire.

– In this study, the author only used IEDB (Vita et al., 2019) and VDJ (Bagaev et al., 2020) databases to prove the detected specificities in their data have been cross–referenced on only five clonotypes. However, the author did not provide experimental evidence showing the pMHC selected by the ATRAP is a real target of a specific clonotype and that those pMHCs removed by the ATRAP are not the target of that clonotype. This may be a rather intensive set of experiments to show this, but the authors could consider it or at least make some statements/caveats if they choose not to do such additional validation.

We agree that further experimental validation is needed to fully confirm the specificities of the TCRs identified/removed by ITRAP. However, we believe it is beyond the scope of the current manuscript to conduct such experiments. As stated earlier, we have however in a separate publication demonstrated that applying ITRAP to denoise SC pMHC-TCR data, results in high predictive power on TCR-specificity predictors compared to training on the raw unfiltered data. We have expanded the discussion to underline this, and clarify that additional validations are needed to fully confirm the identified TCR specificities.

– At lines 654–659, the authors write that low–avidity clonotypes might appear like noise, but this method is only able to detect the binding affinity, not the avidity. The binding affinity of TCR is not always correlated with avidity. I wonder if the information in their dataset really provides avidity measurements.

We agree that the use of the word “avidity” is misleading. We have updated the manuscript and replaced this work with affinity.

– At lines 344–345, it said that "This filtering analysis resulted in optimal thresholds of 2 pMHC UMI counts and a ratio pMHC UMI counts between top one and two >1". Is it possible if the sequence results are deeper that it might result in more noise or background pMHC UMI counts, in such a case, how would one adjust the optimal thresholds?

It is clear that the optimal filtering thresholds are somewhat dependent on the setup applied in the given experiment (see further above). This setup includes the sequencing approach and depth. However, as described in the responses to reviewer 2, one of the powers of ITRAP is the step where the optimal filtering thresholds are defined. The step makes the method flexible with respect to such experimental variations. We have demonstrated this in a benchmark study based on the publicly available 10x data, where we applied ITRAP for denoising of the data. Here slightly different filtering threshold values were identified reflecting the differences in the experimental setup between the two studies. We have included a section to the revised manuscript briefly summarizing these results.

– In general, I think readers would find the V and J gene usage, along with other immune repertoire information interesting for all pMHC binders. The authors should consider this, perhaps as supplementary data.

While this is a very interesting suggestion, we believe such an analysis will add limited value to the current study (where denoising is the prime focus). We have nonetheless made the analysis from the data included in figure 6C (i.e. filtered by UMI thresholds, HLA match, and complete TCRs). The result is shown in Figure 6—figure supplement 1. Here, the gene usage for each positive pMHC multimer is displayed as a Sankey diagram for the 10 most frequent gene-pairs. We suggest including such analyses in future studies where a biological interpretation of the denoised data is of interest.